# *In vivo* study on the repair of rat Achilles tendon injury treated with non-thermal atmospheric-pressure helium microplasma jet

Katusmasa Nakazawa[1], Hiromitsu Toyoda[1,2]*, Tomoya Manaka[2], Kumi Orita[2], Yoshihiro Hirakawa[3], Kosuke Saito[1], Ryosuke Iio[1], Akiyoshi Shimatani[4], Yoshitaka Ban[1], Hana Yao[2], Ryosuke Otsuki[5], Yamato Torii[5], Jun-Seok Oh[5], Tatsuru Shirafuji[5], Hiroaki Nakamura[1,2]

1 Department of Orthopedic Surgery, Graduate School of Medicine, Osaka City University, Abeno, Osaka, Japan, 2 Department of Orthopedic Surgery, Graduate School of Medicine, Osaka Metropolitan University, Abeno, Osaka, Japan, 3 Ishikiriseiki Hospital, Higashiosaka, Osaka, Japan, 4 Saiseikai Nakatsu Hospital, Kita, Osaka, Japan, 5 Department of Physics and Electronics, Graduate School of Engineering, Osaka Metropolitan University, Sumiyoshi, Osakas, Japan

* h-toyoda@omu.ac.jp

## Abstract

Non-thermal atmospheric-pressure plasma (NTAPP) has been widely studied for clinical applications, e.g., disinfection, wound healing, cancer therapy, hemostasis, and bone regeneration. It is being revealed that the physical and chemical actions of plasma have enabled these clinical applications. Based on our previous report regarding plasma-stimulated bone regeneration, this study focused on Achilles tendon repair by NTAPP. This is the first study to reveal that exposure to NTAPP can accelerate Achilles tendon repair using a well-established Achilles tendon injury rat model. Histological evaluation using the Stoll's and histological scores showed a significant improvement at 2 and 4 weeks, with type I collagen content being substantial at the early time point of 2 weeks post-surgery. Notably, the replacement of type III collagen with type I collagen occurred more frequently in the plasma-treated groups at the early stage of repair. Tensile strength test results showed that the maximum breaking strength in the plasma-treated group at two weeks was significantly higher than that in the untreated group. Overall, our results indicate that a single event of NTAPP treatment during the surgery can contribute to an early recovery of an injured tendon.

## Introduction

Tendon is a non-distensible fibrous cord that connects the muscle to the bone or other structures. The Achilles tendon, also known as the calcaneal tendon, is the largest and most biomechanically strong tendon in the human body. It is also one of the most frequently ruptured tendons in our body. A recent Danish study reported an occurrence of Achilles tendon rupture in 31.17/100,000 individuals [1–5]. Conventional treatment for Achilles tendon injuries

**Data Availability Statement:** All relevant data are within the manuscript and its Supporting Information files.

**Funding:** The authors are highly grateful to the Glocal hub of wisdom and wellness filled with smiles for their support. The funders had no role in study design, data collection and analysis, decision to publish, or preparation of the manuscript.

**Competing interests:** The authors have declared that no competing interests exist.

involves rest and immobilization using a brace if the injury is mild; severe injuries require surgical treatments such as suturing or reconstruction. Although surgical treatment can more reliably repair the injured tendon lesion, the return-to-sport rate of patients that undergo surgery is only 60%–80%, with an average return time of 7.6 months and a re-tear rate of 1.7%–5.4% [6–8]. Therefore, the development of adjuvant therapies such as platelet-rich plasma (the proteinaceous fluid portion of the circulating blood) therapy, which can ensure a more reliable and faster return to sport, is underway in many fields [9,10]. However, the effectiveness of such adjuvant therapies is controversial [11,12].

In recent years, non-thermal atmospheric-pressure plasma (NTAPP) has been widely studied for fundamental research and is positively urged for clinical trials [13]. The last two decades has seen a wide application of NTAPP e.g., disinfection, wound healing, cancer therapy, hemostasis, and bone regeneration. NTAPP has also been applied to soft tissues and was found to accelerate skin wound healing and increase the levels of growth factors involved in wound healing [14]. NTAPP irradiation increased the proliferation and migration of human fibroblast primary cells *in vitro* [15]. NTAPP reportedly improves the differentiation potential of osteoblasts. This may be attributed to its potential electrical and chemical effects, such as strong electric field and reactive species or their combination. Moreover, a medical device generating NTAPP, has been clinically used for treating chronic wounds, such as diabetic ulcers [16–18].

NTAPP therapy has been developed since Okazaki found a helium (He) glow discharge at atmospheric pressure. He plasma jet, an NTAPP, has been shown to effectively generate various reactive species, such as electrons, ions, neutral radicals, and photons [19,20]. The He plasma jet can successfully deliver reactive oxygen species (ROS) and reactive nitrogen species (RNS), collectively reactive oxygen and nitrogen species (RONS), to the biological targets, including the skin surface, blood, and tumor, bypassing the biological barriers. We previously demonstrated that NTAPP enhances the repair of injured bones with critical bone defects in an animal model that is unable to heal; He plasma jet irradiation facilitated early filling of the critical bone defects with new bone [21,22]. Therefore, we further investigated the effect of NTAPP treatment on the musculoskeletal system, including bones, cartilage, tendons, and muscles, focusing on the effect of He-plasma-jet-generated RONS on tissue regeneration. As tendon regeneration has been mainly studied *in vitro* using healing effector cells, such as tenocytes and fibroblasts [23–25], we focused on the repair of Achilles tendon. This study aimed to investigate the effects of NTAPP irradiation on injured Achilles tendon in a rat model and histologically and mechanically evaluate the results. We hypothesize that NTAPP irradiation of the Achilles tendon enhances the cell proliferative capacity and promote early Achilles tendon repair.

## Materials and methods

### Helium microplasma jet treatment

A helium microplasma jet was employed as an NTAPP in this study as described previously [20,26]. The device consisted of a 150 mm-long borosilicate glass tube (Pyrex) tapered from an inner diameter of 4 mm to 680 μm at the nozzle as shown in Fig 1A. A 15 mm-long metallic external ring electrode was wound onto the glass tube at a distance of 50 mm from the end of the nozzle. He gas (99.98%, industrial grade) was fed into the glass tube at a fixed gas flow rate of 2.0 L/min. A capillary dielectric barrier discharge microplasma was generated using a sinusoidal high voltage of 10 $kV_{P-P}$ (peak-to-peak) applied to the external electrode at a fixed frequency of 33 kHz, as shown in Fig 1B. This particular plasma jet configuration and operational conditions were chosen because they produced a relatively stable plasma jet (exiting the nozzle) with a plume length of 10 mm and a gas temperature of approximately 40˚C, as

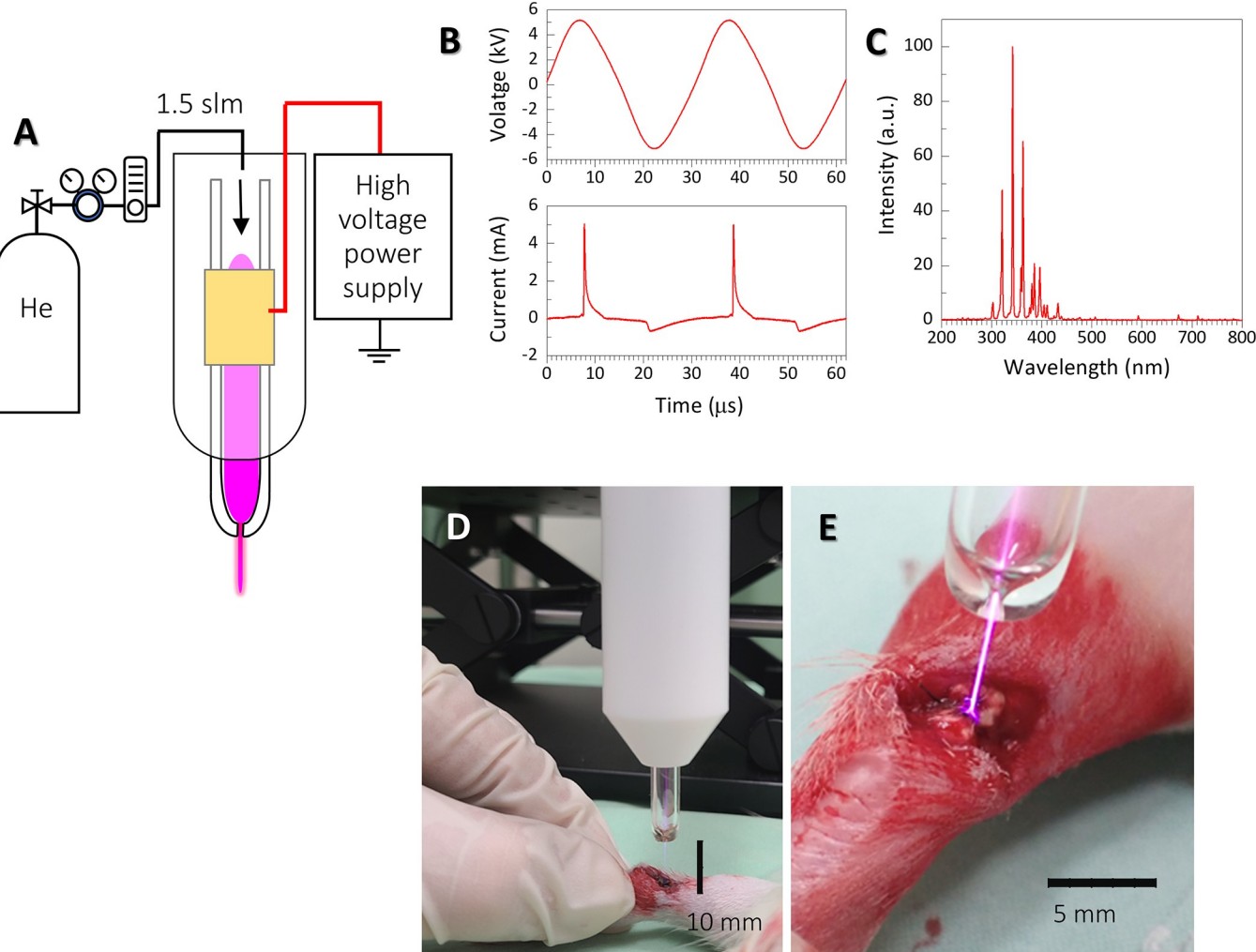

**Fig 1.** (A) The He microplasma jet apparatus used in this study. A stable microplasma jet was generated by He gas through a glass tube with inner diameter tapered from 4 mm to 680 μm at the nozzle exit. (B) AC high voltage and discharge current (10 kV peak to peak at a frequency of 33 kHz) applied to an external ring electrode wound onto the glass tube. (D and E) He microplasma jet treatment for the Achilles tendon injury after surgery.

determined by optical emission analysis of the $N_2$ second positive system (Fig 1C) [20]. Plasma treatments were carried out at a fixed distance of approximately 7 mm between the nozzle and the injured Achilles tendon, as shown in Fig 1D and 1E. Treatment period was set at 3 min, which is relatively shorter than that in our previous study, considering the size of the animal model and the absence of any biological barrier. A preliminary experiment confirmed that the treatment period was sufficient for the Achilles tendon surface to become hydrophilic.

## Animals and surgery procedure

Eight-week-old male Sprague–Dawley rats (n = 36) weighing 260–300 g were used in the experiment. Rats were housed under a 24-h light-dark cycle, with food and water available *ad libitum*. All experimental animal procedures were approved and conducted in accordance with the regulations of the Osaka City University (currently Osaka Metropolitan University) Graduate School of Medicine Committee on Animal Research (approval number: 21085). All efforts were made to minimize suffering. Rats were anesthetized via subcutaneous injection of

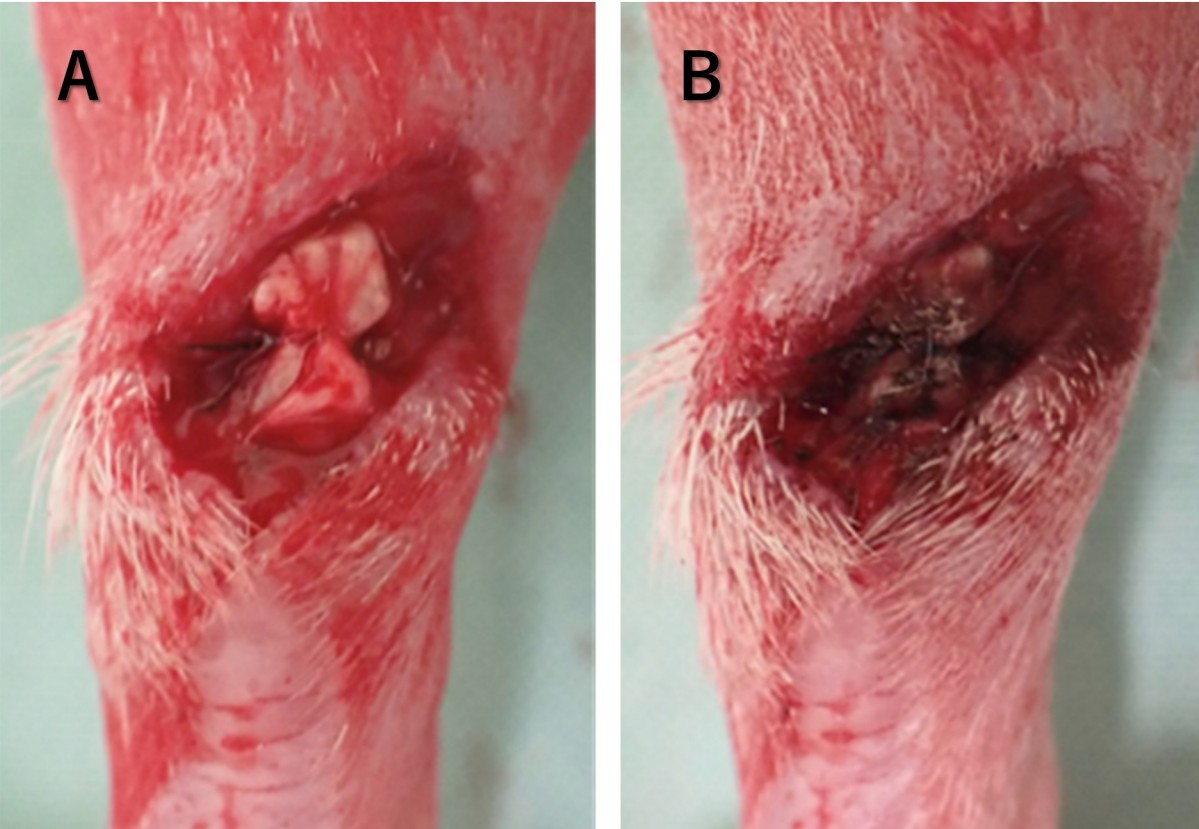

**Fig 2.** (A) The surgery of the Achilles tendon injury rat model. (B) Achilles tendon after He microplasma jet treatment. Hemostasis was confirmed after 3 min of plasma treatment (Fig 2B).

ketamine (50 mg/mL, Daiichi Sankyo, Tokyo, Japan) and xylazine (0.2 mg/mL, Bayer Health-Care Pharmaceuticals, New Jersey, USA) at a ratio of 10:3 and a dose of 1 mL/kg body weight. The posterior part of the foot was shaved using an electric razor. The skin was incised, and the Achilles tendon was separated 5 mm away from the calcaneal attachment. It was then sutured using a modified Kessler technique and a 6–0 nylon monofilament suture as shown in Fig 2A. The Achilles tendon suture was irradiated with the He microplasma jet and hemostasis was confirmed after a 3 min-treatment period (Fig 2B). Rats were weighed weekly and were monitored daily for potential signs of dehydration, pain, infection, and deviant behavior. Rats were euthanized with $CO_2$. at 2, 4 and 6 weeks. All procedures were conducted in compliance with the Animal Research: Reporting of In Vivo Experiments guidelines. No rats met the euthanasia criteria before the experiment.

### Histological testing

The bilateral Achilles tendons and the calcaneus were harvested and fixed overnight in 4% paraformaldehyde (Wako Pure Chemical Industries, Ltd., Osaka, Japan). The specimens were then decalcified in 0.5 M ethylenediaminetetraacetic acid (EDTA) for 1 week and embedded in paraffin. The tissues were cut into 4-μm coronal sections, mounted on slides, deparaffinized with xylene, and dehydrated using a graded ethanol series (70%, 80%, 90%, and 100%). The sections were then stained with hematoxylin and eosin (H&E), Masson's trichrome stain, and Alcian blue, by following standard protocols [27]. The resulting images were analyzed using

CellSens software (Olympus, Tokyo, Japan). Histological analysis was performed by three authors (K.S., Y.H., and H.T.) in a blinded manner. The samples were evaluated using 12 histo-pathological parameters according to Stoll et al. [28], and the histological score was calculated (Table 1). The organizational assessment of the following items, as described by Stoll et al., was performed: the extracellular matrix (ECM) organization of the whole repaired tendon, cellularity, cell alignment, organization of the tendon callus, integration of the constructs into the normal tissue, vascularization, metaplasia, and inflammation features.

Tenocytes are derived from tenoblasts, which represent approximately 90%–to 95% of the tendon fibroblasts. Tenoblasts are round cells with large, oval nuclei, while mature tenocytes are spindle-shaped, long, and thin. Wavy, compact, and parallel-arranged collagen fibers, uni-axial cell alignment, homogeneous cell distribution, and predominantly elongated heterochromatic cell nuclei (tenocytes) are characteristic histological features of tendon tissues. The tissues were scored 0–20 points, with higher scores indicating a more mature tissue repair. Histological evaluation was performed by three investigators (n = 6) blinded to the treatment and number of weeks.

Tissue samples were also stained with picrosirius red (Picro-Sirius Red Stain Kit; ScyTek Laboratories, UT, USA) according to the manufacturer's protocol [29]. The Olympus BX-51Ⓡ microscope with a U-POT (polarizer of transmitted light; Olympus) filter and a U-ANT (analyzer of transmitted light; Olympus) filter as the second filter was used for evaluation. The image was rotated relative to the polarization plane and the digital image was captured at the point of maximum brightness.

The fibers that were strongly birefringent and yellow or red were considered type I collagen, and those that were weakly birefringent and greenish were considered type III collagen, according to Junqueira et al [29]. Based on the report by Su et al., the images were imported into the Image J software (National Institutes of Health, USA) and 8-bit digitized. The non-collagenous fiber tissue was depicted in the dark (0 bit) and the collagenous fiber tissue was depicted on a gray scale from 1 to 255 bits [30]. Three regions (100 × 100 μm) in the site of view were randomly selected from the tendon region of the Achilles tendon suture and evaluated (n = 6). Grayscale values were calculated from the average of these regions and, measured and recorded as previously reported [30], with higher values indicating a higher type I collagen content.

## Immunohistochemical testing

Paraffin-embedded sections were deparaffinized using xylene and dehydrated through graded alcohols. Slides were pretreated with citrate buffer (Target Retrieval Solution [S1699], 10×; DAKO Japan, Tokyo, Japan) in phosphate-buffered saline solution (PBS) for 20 minutes at 90°C for optimal antigen retrieval. Endogenous peroxidases were quenched using 1.0% hydrogen per-oxidase in methanol for 30 minutes at room temperature. Slides were then rinsed with PBS and incubated with 10% goat serum for 30 minutes at room temperature. Subsequently, specimens were incubated with rabbit primary antibodies (type I collagen, 1:100 dilution; Abcam, ab34710 and type III collagen, 1:100 dilution; Abcam, ab6310) at 4°C overnight. After extensive washing with PBS, slides were incubated with a peroxidase-labeled antibody (Histofine Simple Stain; Nichirei Biosciences, Tokyo, Japan) for 30 minutes at room temperature. After extensive washing with PBS, the immunoreaction was visualized by incubating the sections for 3 minutes in 3,3′-diaminobenzidine (Histofine Simple DAB solution; Nichirei Biosciences).

## Biomechanical testing

Six rats each at 2, 4, and 6 weeks post-surgery were used for mechanistic testing. All tests were performed on the same day as the sacrifice. The tendon–muscle junction was fixed proximally

**Table 1. Stoll's histological score.**

| | Points |
|---|---|
| 1: Extracellular matrix (ECM) organization of the whole tendon | |
| Wavy, compact and parallel arranged collagen fibers | 2 |
| In part compact, in part loose or not orderly | 1 |
| Loosely composed, not orderly | 0 |
| 2: Proteoglycan content by Alcian blue staining | |
| Normal | 1 |
| Focally increased | 0 |
| 3: Cellularity/cell-matrix-ratio | |
| Physiological | 2 |
| Locally increased cell density | 1 |
| Increased cell density or decreased ECM content | 0 |
| 4: Cell alignment | |
| Uniaxial | 2 |
| Areas of irregularly arranged cells (10–50%) | 1 |
| More than 50% of cells with no uniaxial alignment | 0 |
| 5: Cell distribution | |
| Homogeneous, physiological | 1 |
| Focal areas of elevated cell density (cell clustering) | 0 |
| 6: Cell nucleus morphology | |
| Predominantly elongated, heterochromatic cell nuclei (tenocytes) | 2 |
| 10–30% of the cells possess large, oval, euchromatic or polymorph heterochromatic nuclei | 1 |
| Predominantly larger, oval, euchromatic or polymorph, heterochromatic nuclei | 0 |
| 7: Organization of repair tissue of the tendon callus | |
| Homogeneous (whole tissue with similar composition) | 2 |
| Locally heterogeneous tissue composition | 1 |
| Whole tissue composition completely changed | 0 |
| 8: Transition from defect to normal tissue | |
| Scaffold integrated, no gaps at the margin visible | 2 |
| Recognizable transition | 1 |
| Abrupt transition, splitting/gaps detectable, callus tissue | 0 |
| 9: Configuration of callus | |
| Normal, only in the defect area, locally confined | 1 |
| Strong, change of whole tendon, thickened | 0 |
| 10: Degenerative changes/tissue metaplasia | |
| Non existing | 3 |
| Moderate formation of oedema | 2 |
| Intense oedema with inclusion of fat, cell and/or fibres destruction, fibrin deposition, gaps" | 1 |
| Assembly of cartilage or bone (alcian blue staining) | 0 |
| 11: Vascularisation in the defect area | |
| Hypo-vascularized, like surrounding tendon (small capillaries) | 1 |
| Hyper-vascularized (increased numbers of small or larger capillaries) | 0 |
| 12: Inflammation | |
| No inflammatory cell infiltrates | 1 |
| Infiltrating inflammatory cell types (neutrophils, macrophages, foreign-body/giant cell) | 0 |

and distally to the repair site using sandpaper and glue, and the calcaneus and muscle belly at the tendon origin were fixed with clamps, as reported by Kennedy *et al* [31]. The tendon was loaded at 1 cm/min until rupture. The ultimate load to failure was then measured using an apparatus (Shimadzu Corp., Kyoto, Japan) and recorded.

### X-ray photoelectron spectroscopy (XPS) and surface wettability

To identify the surface chemistry and wettability changes, XPS was performed and the water contact angle (WCA) was measured. XPS was performed on-site using a small piece of Achilles tendon, according to previous reports [21]. The untreated Achilles tendons were assessed using an XPS instrument (ESCA-3400, Shimadzu, Kyoto, Japan). Subsequently, the same procedure was used to assess each treated (irradiated with He microplasma jet for 3 min) Achilles tendon. Furthermore, the WCA was measured using a contact angle analyzer (DMe-211, Kyowa Interface Sci. Co., Saitama, Japan) and a protocol reported previously was followed. Deionized water (1 μL) was dropped onto a section of each Achilles tendon, either untreated or plasma-treated (Fig 3).

### Statistical analysis

Statistical analyses were performed using SPSS (version 25.0, Microsoft Windows, SPSS, Chicago, IL, US). Data are expressed as the mean value with standard deviation. Data were analyzed using the Mann-Whitney U test. Statistical significance was set at $p < 0.05$. Interobserver agreement of the Stoll's histological score between the three authors was assessed using intraclass correlation coefficients (0.000–0.200, poor; 0.201–0.400, fair; 0.301–0.600, moderate; 0.601–0.800, good; and 0.801–1.000, very good).

## Results

### Histological analysis

The ICC of Stoll's histological score was 0.861, which is considered excellent. Images of the respective tissues are shown in Figs 4–7.

In the plasma-treated group, elongated cells were observed in the 2-week histological evaluation; at 4 weeks, most cells were elongated, and fibrocartilage tissue emerged. The tissue arrangement became more regular after 2 weeks and matured over time. In contrast, in the untreated group, nuclear accumulation was observed at 2 weeks and the cells were predominantly oval. The tissue arrangement became more regular over time; however, a cluster of cells was still observed at 6 weeks, with fewer elongated cells being present compared to the amount found in the plasma-treated group. Similar to what was found in the plasma-treated group, fibrocartilaginous tissue was observed at 4 weeks (Fig 4). Thus, NTAPP irradiation resulted in a more regular tissue arrangement and an earlier appearance of tenocyte-like cells.

The Stoll's histological score results showed that the plasma-treated group had 10.3 (1.5) points at 2 weeks, 11.3 (2.0) points at 4 weeks, and 11.3 (1.9) points at 6 weeks. The Stoll's histological score in the untreated group was 5.9 (1.0) points at 2 weeks, 7.8 (0.9) points at 4 weeks, and 9.3 (1.9) points at 6 weeks. The comparison of the scores at each time point showed a significantly higher score in the plasma-treated group at 2 ($p = 0.003$) and 4 weeks ($p = 0.006$); however, no significant difference was observed at 6 weeks ($p = 0.164$).

In the time-point comparison of Stoll's histological scores, that of the plasma-treated group reached a plateau at 2 weeks, with no significant differences between the scores obtained at 2 and 4 weeks ($p = 0.441$), 4 and 6 weeks ($p = 0.991$), and 2 and 6 weeks post-surgery ($p = 0.672$). In the untreated group, significant differences were found between the scores

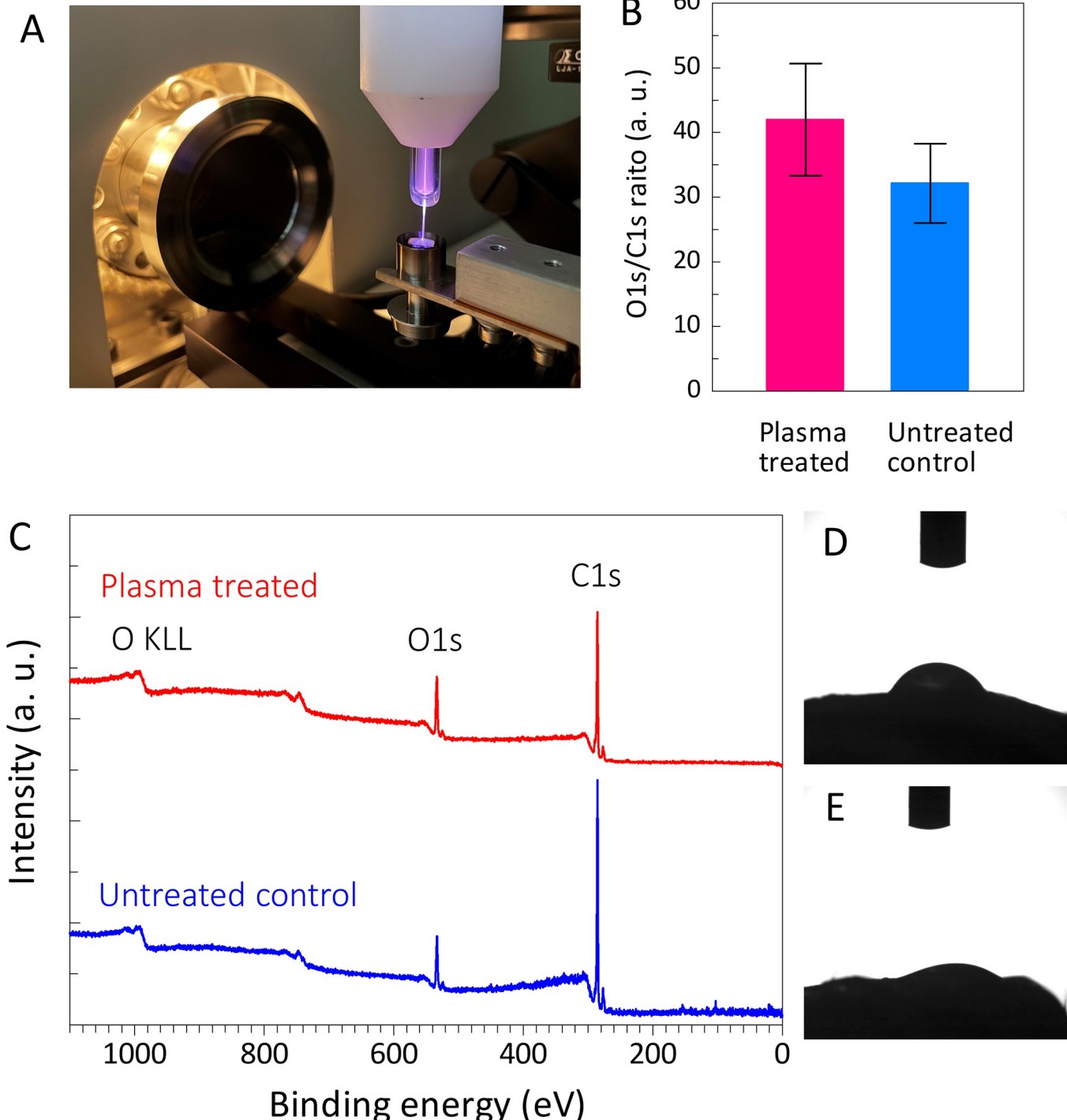

**Fig 3.** (A) Wide-scan XPS spectra measured on-site before and after the plasma treatment. (B and C) A significant increase in the O1s peak intensity and a decrease in the C1s peak intensity were observed after the plasma treatment. (D) WCA results before and after irradiation.

obtained at weeks 2 and 4 (p = 0.032) and weeks 2 and 6 (p = 0.008) post-surgery, and no significant difference was observed between the scores obtained at weeks 4 and 6 post-surgery (p = 0.099) (Fig 8). The results showed that NTAPP irradiation resulted in early tissue maturation.

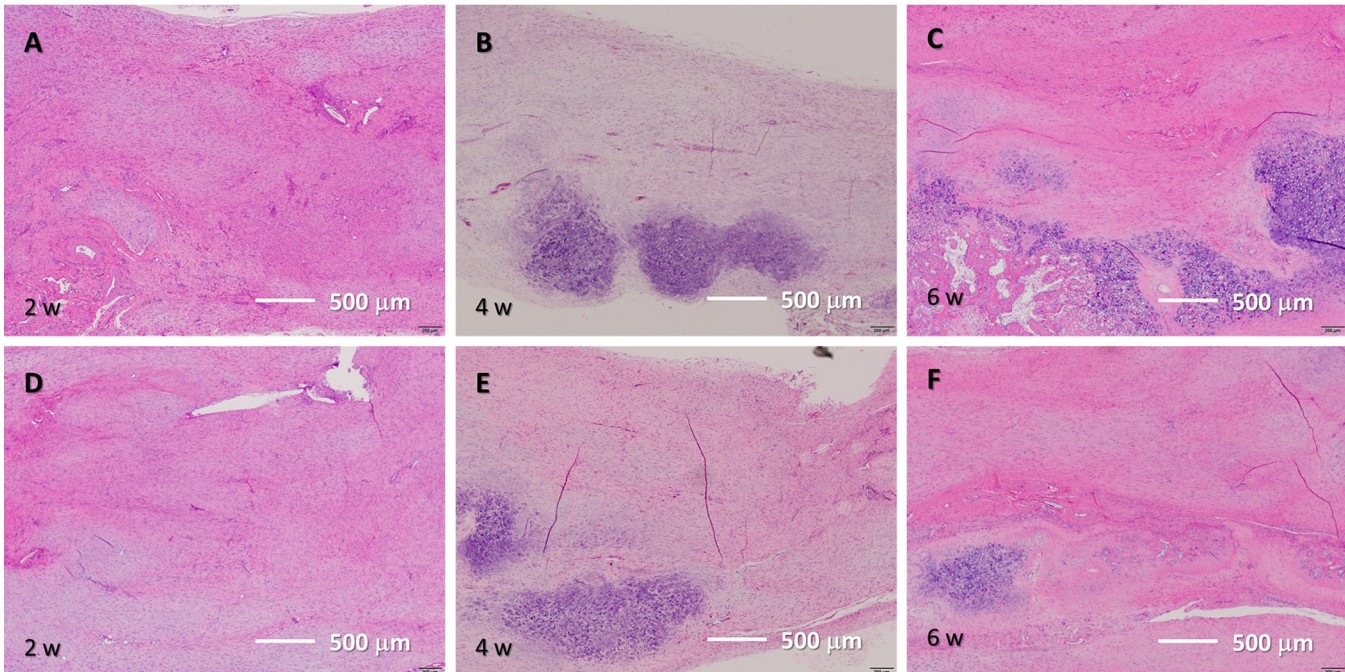

**Fig 4. H&E staining results.** (A–C) Tissue after He microplasma jet treatment at 2, 4, and 6 weeks post-surgery, respectively. (D–F) Untreated control tissue at 2, 4, and 6 weeks post-surgery, respectively. Scale bar, 500 μm (originally 200 μm).

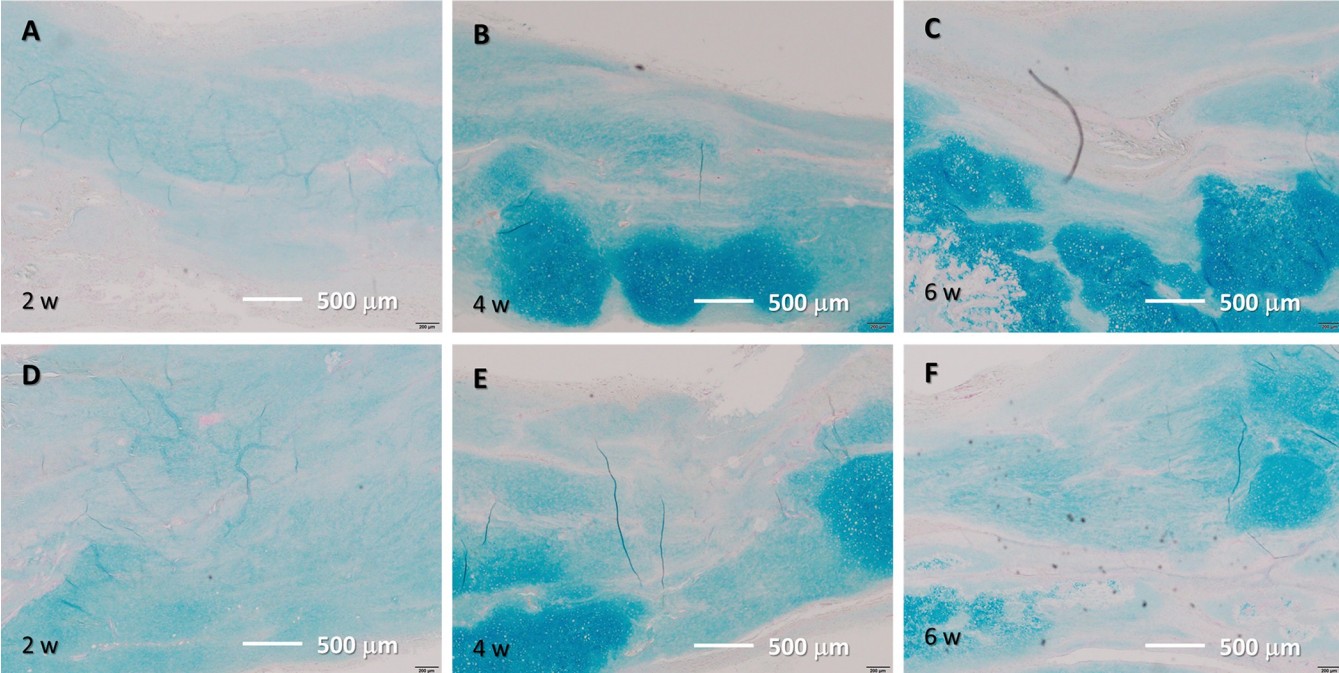

**Fig 5. Alcian blue staining results.** (A–C) Tissue after He microplasma jet treatment at 2, 4, and 6 weeks post-surgery, respectively. (D–F) Untreated control tissue at 2, 4, and 6 weeks post-surgery, respectively. Scale bar, 500 μm (originally 200 μm).

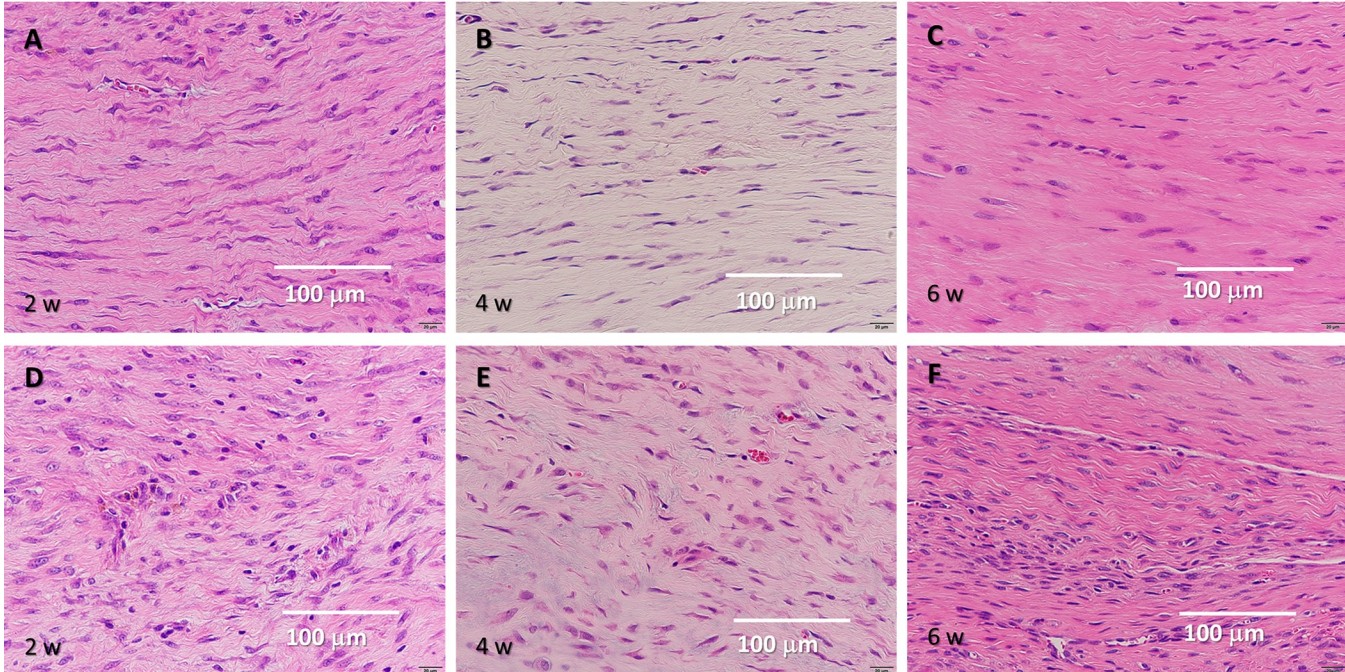

**Fig 6. H&E staining results.** (5–C) Tissue after He microplasma jet treatment at 2, 4, and 6 weeks post-surgery, respectively. (5–F) Untreated control tissue at 2, 4, and 6 weeks post-surgery, respectively. Scale bar, 100 μm (originally 20 μm).

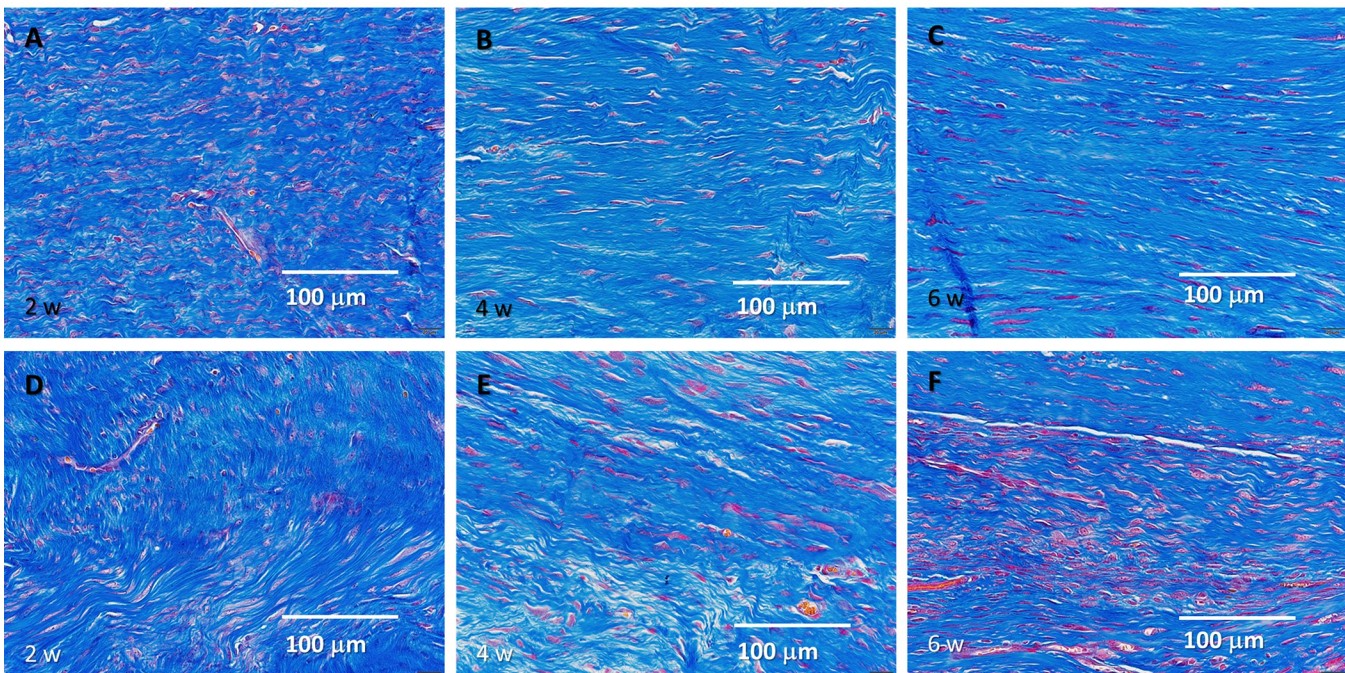

**Fig 7. Masson's trichrome staining results.** (A–C) Tissue after He microplasma jet treatment at 2, 4, and 6 weeks post-surgery, respectively. (6–F) Untreated controls at 2, 4, and 6 weeks post-surgery, respectively. Scale bar, 100 μm (originally 20 μm).

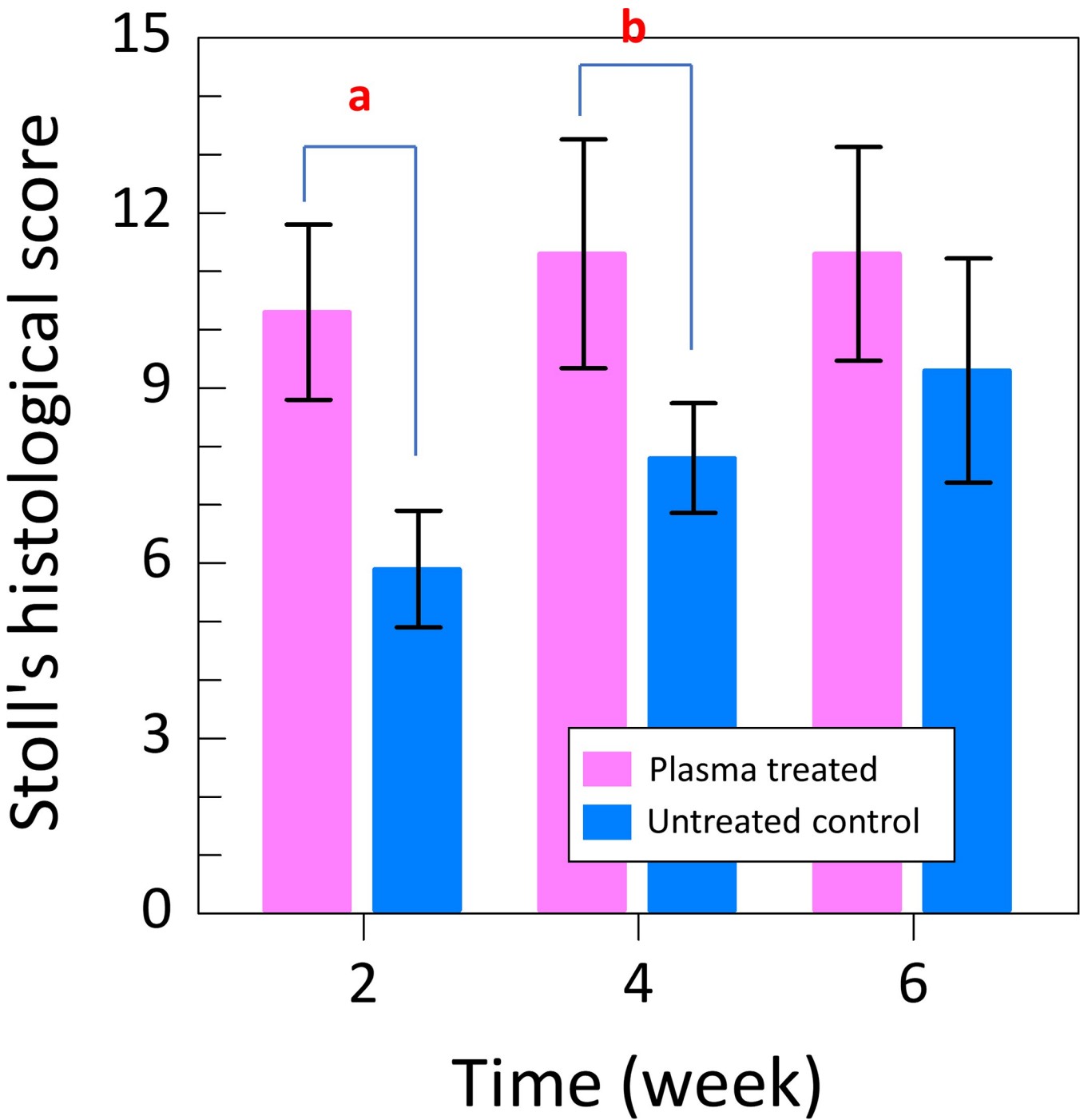

**Fig 8. Stoll's histological scores for He microplasma jet treated and untreated control tissues.** The scores for the He microplasma jet treated tissues was higher than that of the untreated controls.

The item-by-item comparison revealed that there was no significant difference between items at 2 weeks. However, at 4 weeks, significant differences were observed in cell alignment, cell nucleus morphology, and vascularization in the defect area. At 6 weeks, a significant difference occurrent in the transition from defect to normal tissue (Table 2).

**Table 2. Comparison of each Stoll's histological score item (excluding proteoglycan content by Alcian blue staining).**

| 2 weeks | Extracellular matrix (ECM) organization of the whole tendon | Cellularity/cell-matrix-ratio | Cell alignment | Cell distribution | Cell nucleus morphology | |
|---|---|---|---|---|---|---|
| Untreated control | 1.2±0.4 | 1.0±0 | 0.5±0.5 | 0±0 | 0.5±0.5 | |
| Plasma treated | 1.0±0 | 1±0 | 0.5±0.5 | 0±0 | 0.7±0.8 | |
| P Value | 0.341 | 1 | 1 | 1 | 0.687 | |
| 2 weeks | Organization of repair tissue of the tendon callus | Transition from defect to normal tissue | Configuration of callus | Degenerative changes/tissue metaplasia | Vascularisation in the defect area | Inflammation |
| Untreated control | 1.2±0.4 | 1.6±0.5 | 0±0 | 1.3±0.5 | 0.7±0.5 | 0.2±0.4 |
| Plasma treated | 1.2±0.4 | 1.3±0.8 | 0±0 | 1.8±0.8 | 0.8±0.4 | 0.2±0.4 |
| P Value | 1 | 0.418 | 1 | 0.209 | 0.549 | 1 |
| 4weeks | Extracellular matrix (ECM) organization of the whole tendon | Cellularity/cell-matrix-ratio | Cell alignment | Cell distribution | Cell nucleus morphology | |
| Untreated control | 1.0±0.6 | 0.8±0.4 | 1.0±0 | 0±0 | 1.0±0 | |
| Plasma treated | 1.2±0.4 | 1.0±0 | 1.5±0.5 | 0.2±0.4 | 1.5±0.5 | |
| P Value | 0.599 | 0.341 | 0.049 | 0.341 | 0.049 | |
| 4weeks | Organization of repair tissue of the tendon callus | Transition from defect to normal tissue | Configuration of callus | Degenerative changes/tissue metaplasia | Vascularisation in the defect area | Inflammation |
| Untreated control | 1.0±0 | 2.0±0 | 0±0 | 0.2±0.4 | 0.5±0.5 | 0.3±0.5 |
| Plasma treated | 1.3±0.5 | 2.0±0 | 0±0 | 0.7±1.0 | 1.0±0 | 1.0±0 |
| P Value | 0.145 | 1 | 1 | 0.296 | 0.049 | 0.01 |
| 6 weeks | Extracellular matrix (ECM) organization of the whole tendon | Cellularity/cell-matrix-ratio | Cell alignment | Cell distribution | Cell nucleus morphology | |
| Untreated control | 0.8±0.4 | 1.0±0 | 0.5±0.5 | 0.2±0.4 | 0.7±0.5 | |
| Plasma treated | 1.2±0.8 | 1.0±0 | 0.8±0.4 | 0.3±0.5 | 0.8±0.4 | |
| P Value | 0.363 | 1 | 0.26 | 0.549 | 0.549 | |
| 6 weeks | Organization of repair tissue of the tendon callus | Transition from defect to normal tissue | Configuration of callus | Degenerative changes/tissue metaplasia | Vascularisation in the defect area | Inflammation |
| Untreated control | 1.0±0 | 1.5±0.5 | 1.0±0 | 1.0±1.1 | 1.0±0 | 1.0±0 |
| Plasma treated | 1.0±0 | 2±0.1 | 1.0±0 | 1.3±0.6 | 1.0±0 | 1.0±0 |
| P Value | 1 | 0.049 | 1 | 0.599 | 1 | 1 |

## Quantitative analysis of the type I collagen content via picrosirius red staining

Picrosirius red staining of collagen networks in plasma-treated and untreated control tissues are shown in Fig 9. The plasma-treated group (Fig 9A–9C) shows the collagen networks in yellow and red. Interestingly, yellow and red birefringence was most commonly observed in the plasma-treated group at an earlier stage (2 weeks post-surgery). In contrast, greenish birefringence was observed in the untreated control group, even as late as 6 weeks post-surgery.

Quantitative analysis showed that the type I collagen content was higher in plasma-treated group than that in untreated control group at all time points (2, 4, and 6 weeks) post-surgery ($p = 0.001$, $p = 0.038$, and $p = 0.001$, respectively) (Fig 10).

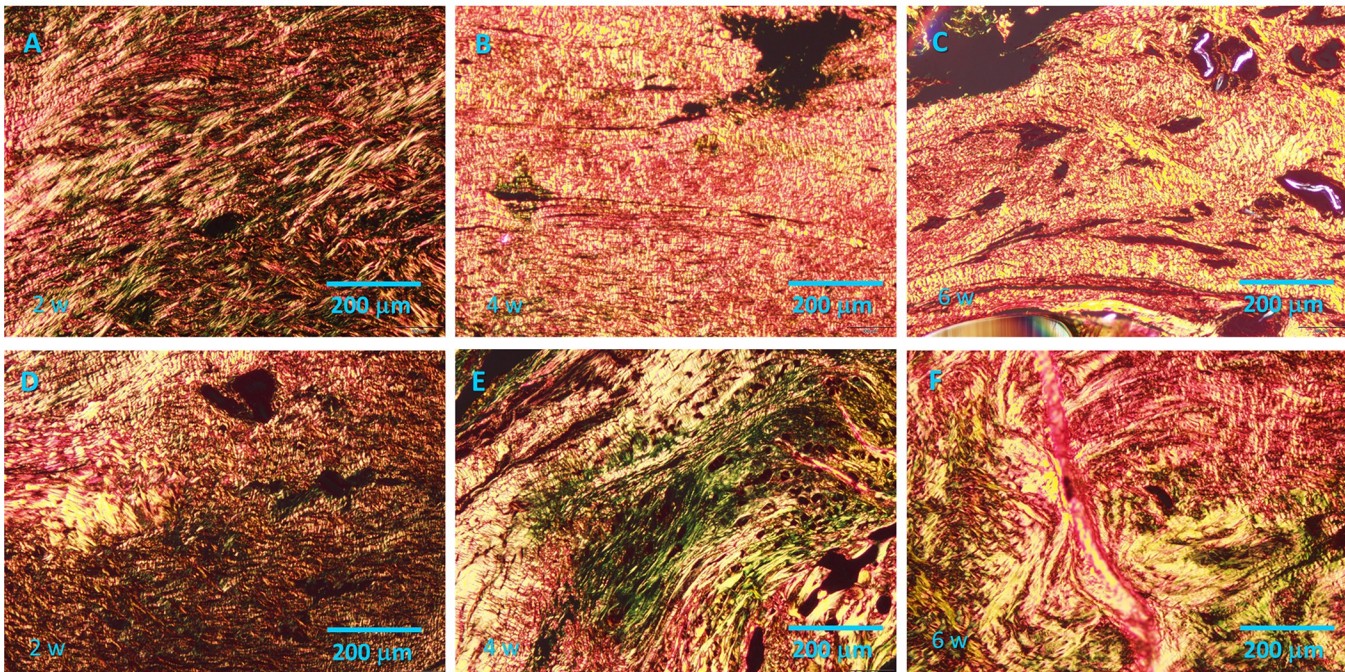

**Fig 9. Picrosirius red staining results.** (A–C) Tissue after He microplasma jet treatment at 2, 4, and 6 weeks post-surgery, respectively. (D–F) Untreated control tissues at 2, 4, and 6 weeks post-surgery, respectively. Scale bar, 200 μm (originally 100 μm).

## Quantitative analysis of the type I and III collagen content via immunohistochemistry testing

Type I collagen expression was observed in the plasma-treated group from 2 weeks and became stronger with time, showing stronger staining compared to the untreated control group during the entire period (Fig 11A–11F). Type III collagen expression was stronger in the untreated control group than in the plasma-treated group at 2 and 4 weeks (Fig 12A, 12B, 12D and 12E), but the staining was similar at 6 weeks (Fig 12C and 12F). Expression decreased over time in both groups. In quantitative analysis, type I collagen expression was significantly higher in the plasma-treated group at 2, 4, and 6 weeks. (p = 0.010, p = 0.025, and p = 0.025, respectively). Type III collagen expression was significantly lower in the plasma-treated group at 2 and 4 weeks (p = 0.010 and p = 0.037), but there was no significant difference between the two groups at 6 weeks. (p = 0.262) (Fig 13).

## Biomechanical testing

Ultimate load to failure, also known as tensile strength, was obtained to confirm the recovery of Achilles tendon after the surgery up to 6 weeks. A significant difference was observed between the plasma-treated and untreated control groups (Fig 14), with higher values of the ultimate load to failure in all samples of plasma-treated group at each time point up to 6 weeks. The value was 67.2 ± 14.4 N and 33.3±15.4 N for plasma-treated and untreated control groups, respectively, after 2 weeks post-surgery. That value of the ultimate load to failure for plasma-treated group at 2-week time point was similar that at 4 week (58.8±20.8 N) and 6 week (77.6±25.5 N) time points. A comparison of these values at each time point revealed a significant improvement in the plasma-treated group at 2 weeks (p = 0.002), and no significant differences at 4 and 6 weeks post-surgery (p = 0.262 and p = 0.337, respectively).

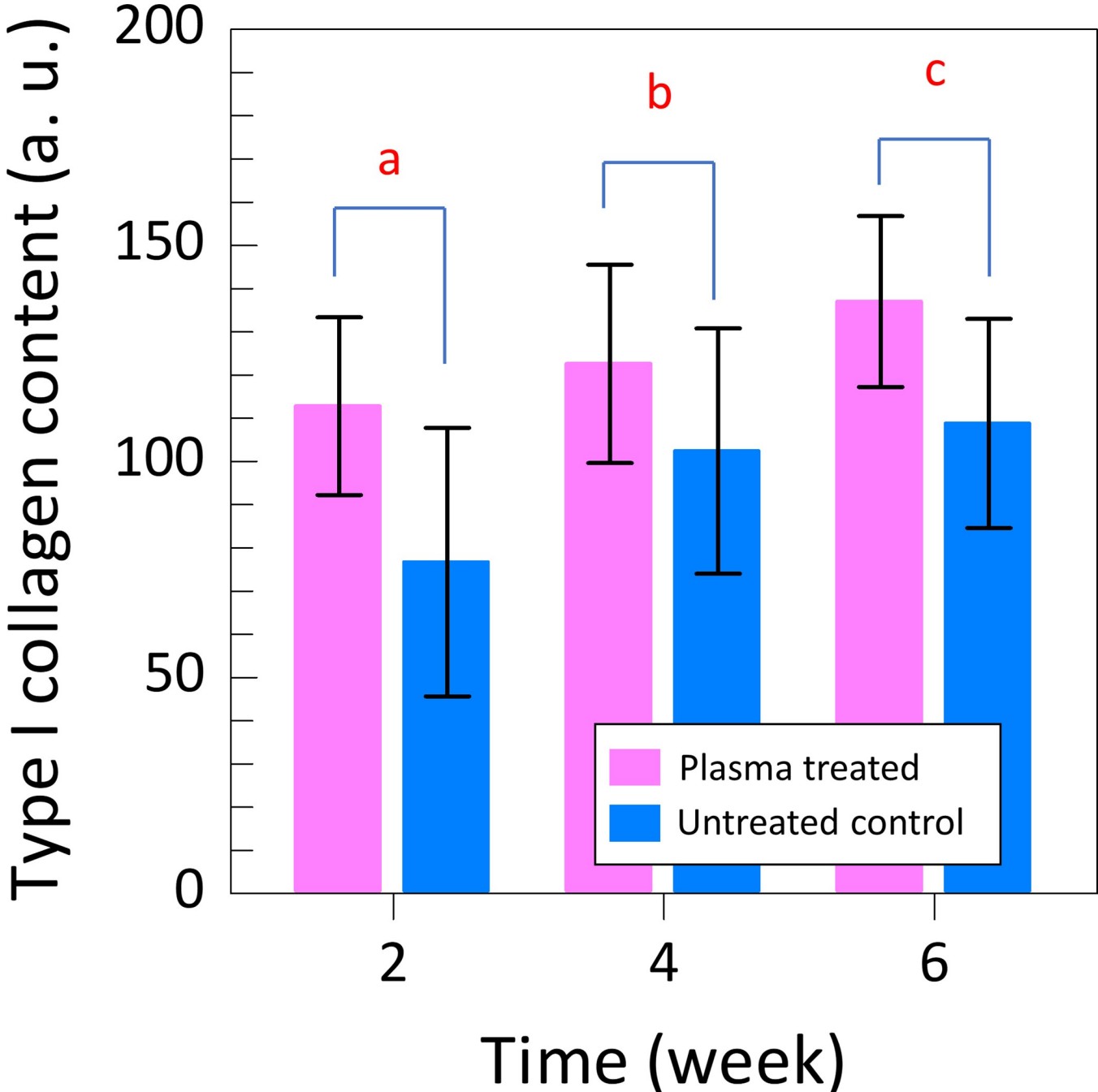

**Fig 10. Quantitative analysis of the type I collagen content via picrosirius red staining.** Type I collagen content in He microplasma jet-treated group was higher than that in untreated control group. (a: p = 0.001, b: p = 0.038 c: p = 0.001).

### XPS and surface wettability

The on-site XPS measurements showed that the O1s peak intensity (at 533 eV) increased and the C1s peak intensity (at 286 eV) decreased after 3 min of plasma irradiation (Fig 3B and 3C). This was considered a result of etching by plasma irradiation, as reported previously [21]. The increase in the O1 peak intensity was associated with the oxidation of the tendon surface by plasma irradiation, which improves cell adhesion and is related to an increased hydrophilicity.

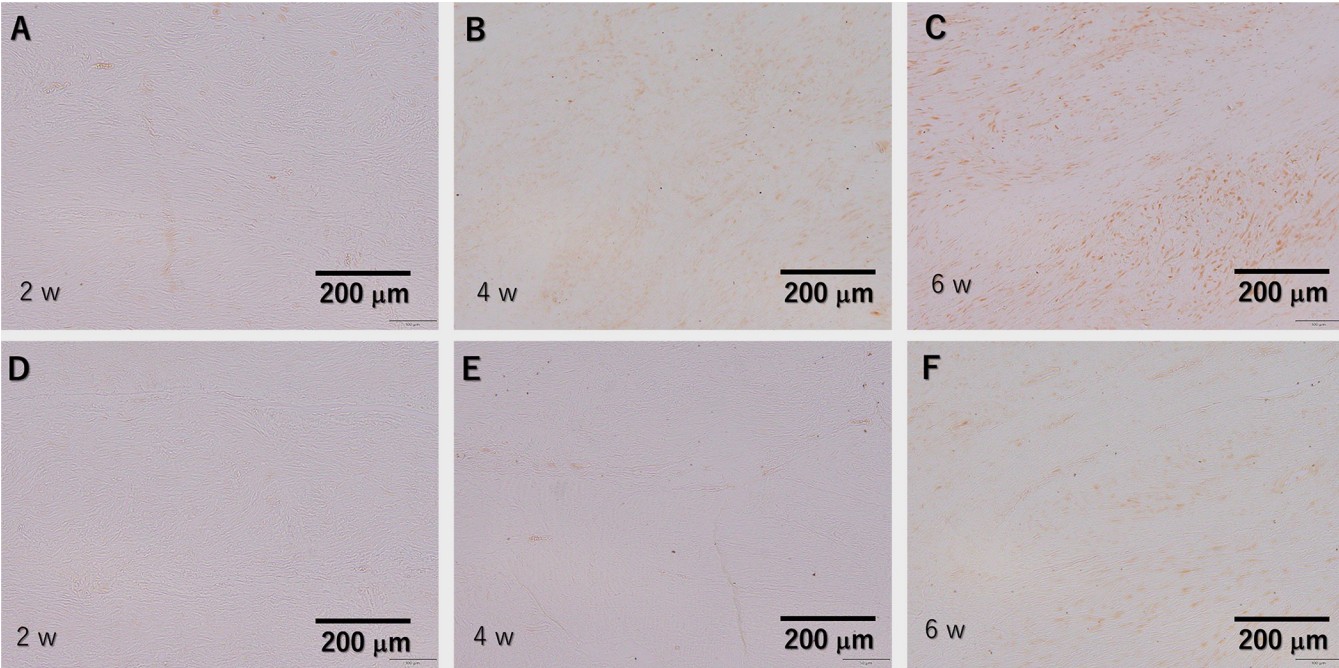

**Fig 11. Immunohistochemical testing against type I collagen results.** (A–C) Tissue after He microplasma jet treatment at 2, 4, and 6 weeks post-surgery, respectively. (D–F) Untreated control tissues at 2, 4, and 6 weeks post-surgery, respectively. Scale bar, 200 μm (originally 100 μm).

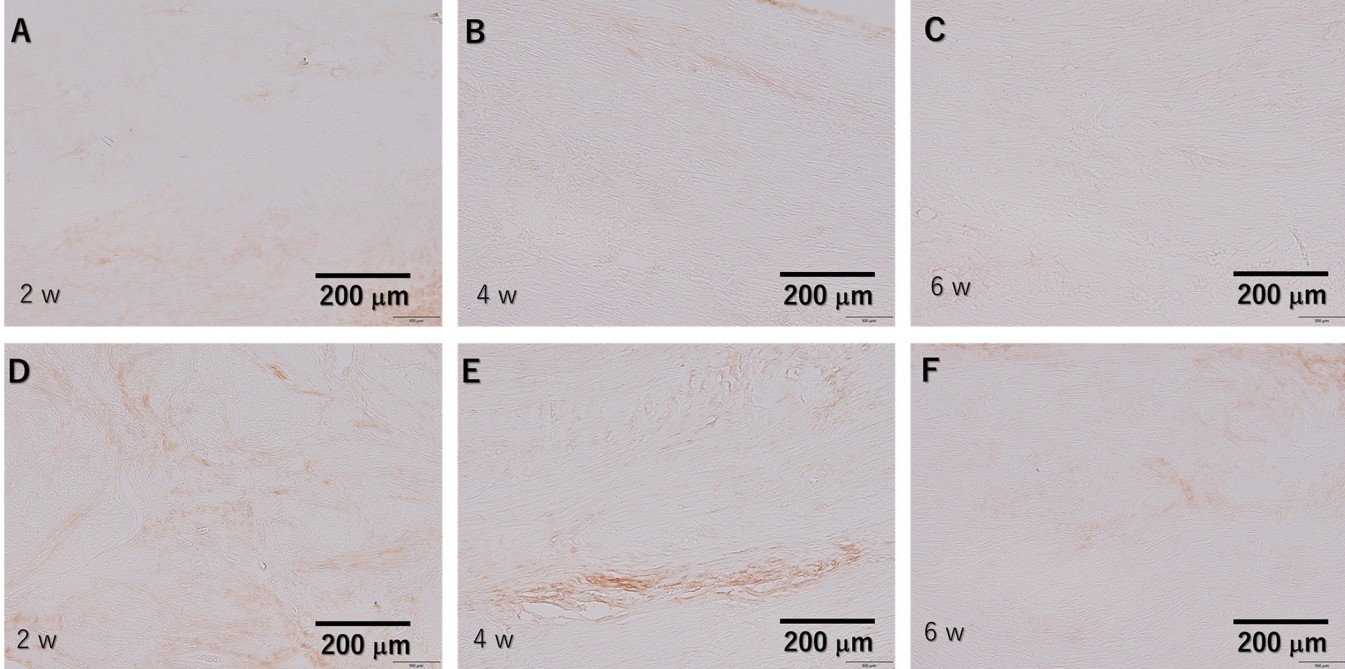

**Fig 12. Immunohistochemical testing against type III collagen results.** (A–C) Tissue after He microplasma jet treatment at 2, 4, and 6 weeks post-surgery, respectively. (D–F) Untreated control tissues at 2, 4, and 6 weeks post-surgery, respectively. Scale bar, 200 μm (originally 100 μm).

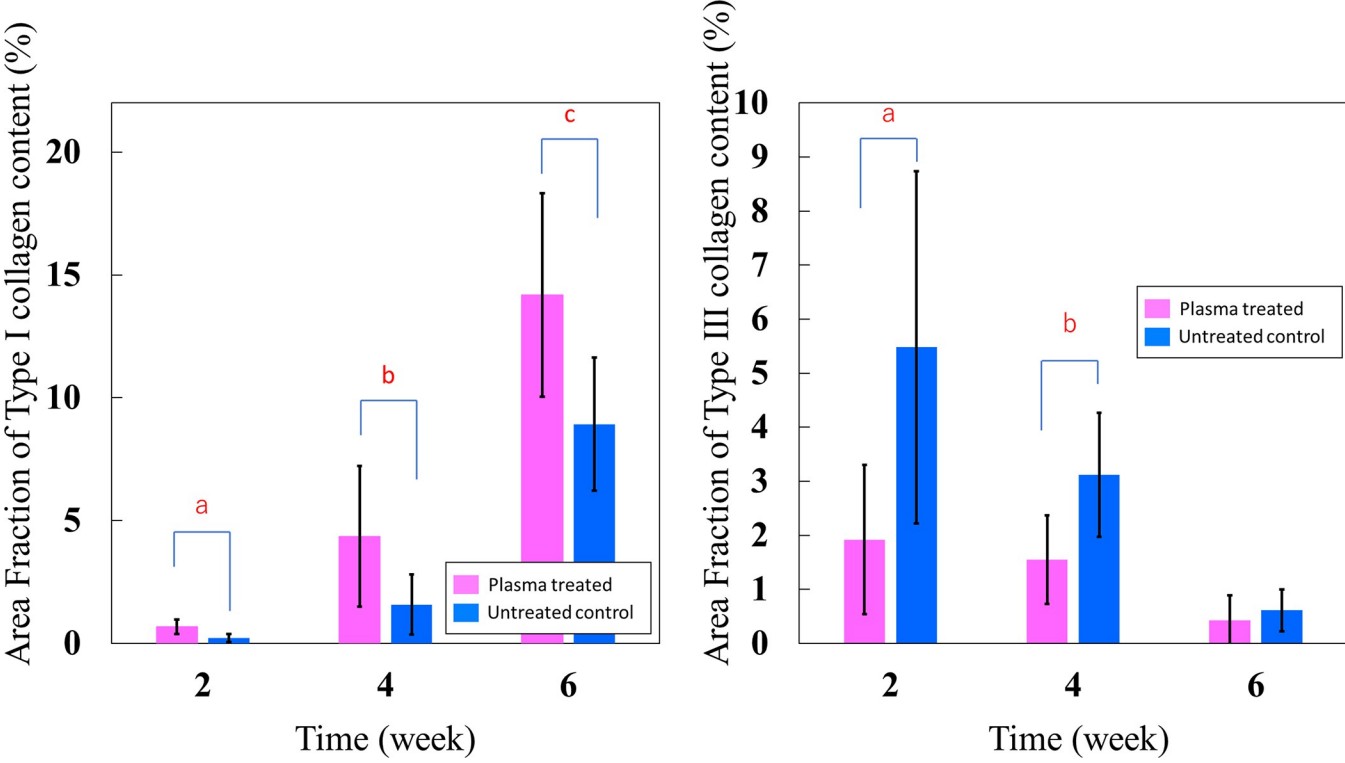

**Fig 13. Quantitative analysis of the type I and III collagen content via immunohistochemistry testing Type I collagen expression in He microplasma jet-treated group was higher than that in untreated control group in all time point.** (a: p = 0.010, b: p = 0.025, c: p = 0.025) Type III collagen expression was significantly lower in the He microplasma jet-treated group at 2 and 4 weeks. (a: p = 0.010, b: p = 0.037).

The WCA of the untreated group was 65.9˚ (3.9˚), as measured using a contact angle analyzer; however, that of the plasma-treated group was only 30.8˚ (0.9˚) (Fig 3D and 3E).

## Discussion

This is the first study to investigate the effect of direct NTAPP irradiation on tendon healing. Here, histological evaluation showed a significant difference in the Stoll's histological score at 2 and 4 weeks post-surgery, and a higher score at 6 weeks post-surgery in the plasma-treated group. The type I collagen content, assessed using picrosirius red staining, was higher in the plasma-treated group at 2, 4, and 6 weeks post-surgery. Immunohistochemistry testing results also showed an increase in type I collagen and a decrease in type III collagen in the plasma-treated group at 2 and 4 weeks. Mechanical evaluation revealed a significantly higher load to failure at 2 weeks post-surgery. Our results suggest that NTAPP irradiation may accelerate Achilles tendon repair.

The results of the *in vivo* experiments showed that NTAPP irradiation resulted in significant differences in the Stoll's histological scores at 2 and 4 weeks post-surgery, with a higher score being observed at 6 weeks post-surgery. Tendon repair involves three different phases: inflammatory, fibroblastic, and remodeling [32–35]. Scar tissue is formed by fibroblasts and inflammatory cells during the inflammatory phase, within 1–2 weeks post-surgery. Most of the collagen produced during this process is type III [36–38]. Subsequently, in the fibroblastic phase, occurring between weeks 3 and 6 post-surgery, fibroblast proliferation and the production of collagen and other ECM components become evident [38,39]. Around week 6 post-surgery, the remodeling phase, in which the new collagen sequences are assembled in parallel,

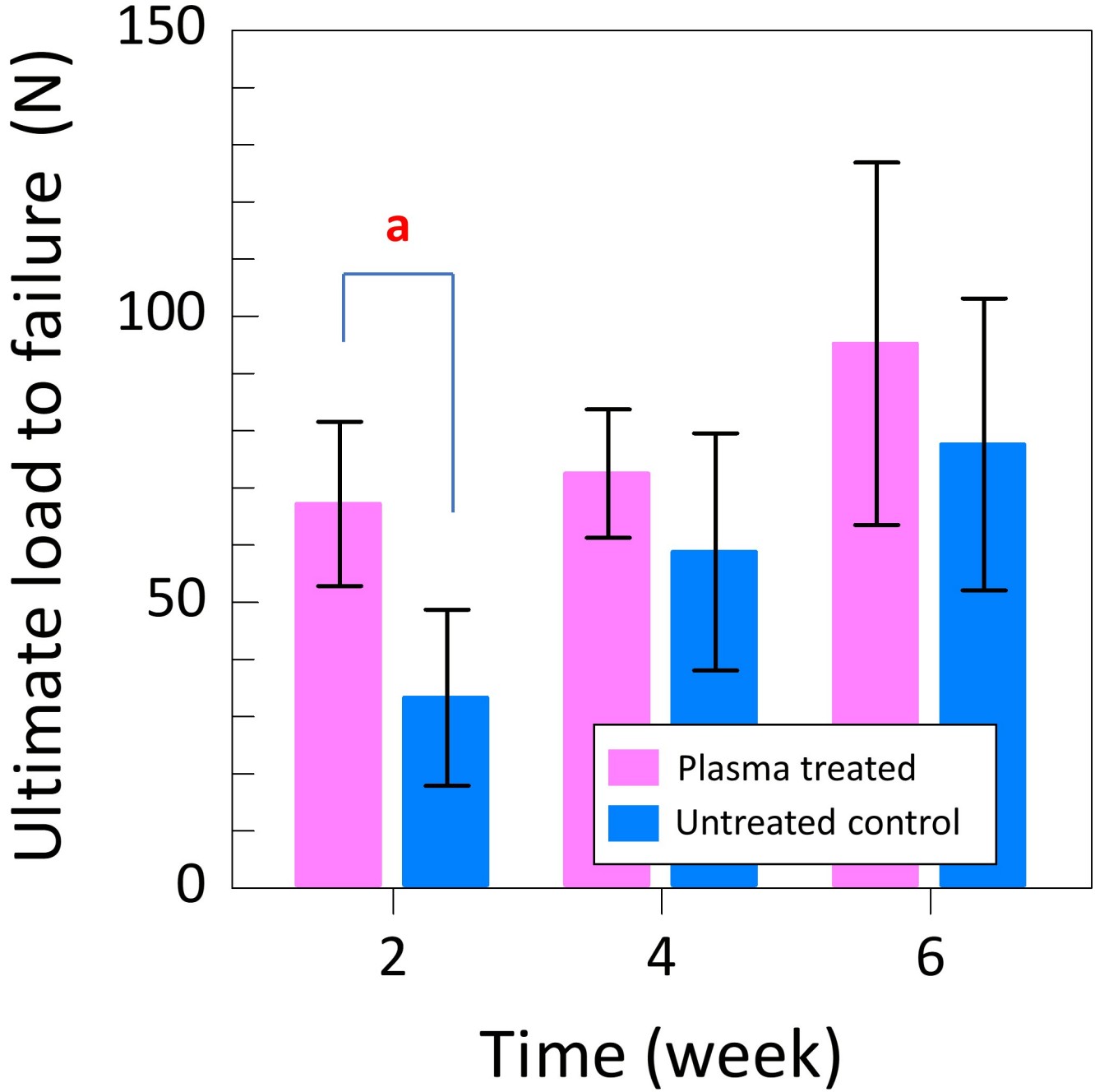

**Fig 14. The results of Biomechanical testing.** At 2 weeks, the ultimate load to failure was significantly higher in the plasma-treated group. (a: p = 0.002).

occurs [33,34,40]. The replacement of collagen type III with type I also results in the tissue becoming more fibrotic, having a decreased cellularity and a lower matrix production [37,41–43].

Here, the untreated group also showed scar tissue formation at approximately 2 weeks post-surgery and parallel-arranged collagen fibers at 6 weeks post-surgery. In contrast, in the plasma-treated group, the parallel-arranged collagen fibers were observed from an early stage, and the Stoll's tissue score at 2 weeks post-surgery was comparable to that at 6 weeks post-

surgery in the untreated group. At 2 weeks post-surgery, the collagen fibers were parallel, and elongated tendon cells appeared from an early stage. No significant difference in the Stoll's tissue histological scores between weeks 2, 4, and 6 post-surgery was observed, and the threshold was reached early, confirming that NTAPP irradiation causes early tissue repair. In addition, cartilage formation was detected starting 2 weeks postoperatively in Alcian blue stainning. These cartilage masses have been reported to calcify after 4 to 6 weeks in previous reports and are thought to be caused by endochondral ossification [44]. Moreover, when comparing each item, a significant difference was observed at 4 weeks in cell alignment, cell nucleus morphology, and vascularization in the defect area. Indeed, at 6 weeks, a significant difference was detected in transition from defect to normal tissue. Collectively, these factors may have contributed to the NTAPP-induced accelerated repair. These results indicate that NTAPP irradiation suppresses scar formation and promotes the regeneration of mature tendons.

Normal Achilles tendon dry weight is composed primarily of collagen type I (90%) [45,46]. However, it also contains type III collagen, present in the fibrocartilage of tendons and tendons, as well as type V collagen, which forms the nuclei type I collagen protofibrils and participates in fiber diameter regulation [47]. Picrosirius red staining detects and quantitatively evaluates collagen in normal and overlying tissue sections and can effectively identify collagen type I and collagen type III in the musculoskeletal system, liver, gastrointestinal tract, skin, and myocardium [48]. Achilles tendon repair can be achieved by replacing collagen type III with type I to obtain cell maturity [43]. In other words, a higher type I collagen content confirms that the Achilles tendon has matured. Here, the type I collagen content also resulted in higher brightness in Picrosirius red staining and the expression of type I collagen in immunochemical testing at 2, 4, and 6 weeks post-surgery. The expression of type III collagen was also decreased in the plasma treatment group at 2 and 4 weeks post-surgery. This result is consistent with the Stoll's histological score results, which showed that collagen replacement occurred at an early stage. Regarding the mechanical testing results, significant differences in the ultimate load to failure were observed at 2 weeks post-surgery, but not at 4 or 6 weeks post-surgery. Regarding the mechanical testing results obtained from the Achilles tendon injuries, early maturation and similar tissue repair results were observed. We consider that the load to failure improved earlier owing to an accelerated tendon repair. Here, fibrocartilage tissue formation was observed in both groups at 4 weeks post-surgery; Rooney et al. reported the appearance of various cartilages in rats from 4 weeks after Achilles tendon transection [49]. The same was observed in the present study.

Here, the hydrophilicity of the Achilles tendon was assessed; after 3 min of irradiation, the WCA was 31.6° in the plasma-treated group and 65.9° in the untreated group, being lower in the former. The on-site XPS measurements showed a significant increase in the O1 peak intensity after the plasma treatment, indicating a moderate oxidation of the tendon surface due to NTAPP irradiation. Therefore, a 3-minute irradiation may improve the hydrophilicity of the Achilles tendon surface, and thus facilitate tendon repair.

NTAPP irradiation has been shown to increase the hydrophilic properties and osteoinductive capacity of porous calcium hydroxyapatite [50]. In our previous report, we found that irradiating bone with NTAPP improved the hydrophilicity of cells [21]. In tendons, an increased hydrophilicity may improve the ability to attract cells, such as tenocytes, into the ECM.

NTAPP produces ROS and RNS, including nitrogen monoxide (NO) and nitrogen dioxide (NO2), by interacting with the surrounding atmosphere when gases are irradiated from the plasma jet [51]. Only recently has the role of ROS in the maintenance of normal biological functions become clear [52]. ROS activate intracellular signaling pathways, resulting in the production of soluble factors involved in cell growth and proliferation [53,54]. According to Brun et al., He-generated plasma treatment induces the proliferation and migration of primary

human fibroblast-like cells, primarily through intracellular ROS formation [15]. Accordingly, we speculate that the RONS generated through the plasma jet may promote tendon regeneration.

Recently, reports on NTAPP irradiation of living organisms have been published. NTAPP reportedly promotes wound healing [14]. The underlying mechanism involves an increase in the levels of growth factors, such as vascular endothelial growth factor (VEGF) and granulocyte-macrophage colony-stimulating factor (GM-CSF) [55]. We have previously reported that NTAPP irradiation can enhance the filling of bone defects [21]. The tumor necrosis factor (TNF), VEGF, platelet-derived growth factor, connective tissue growth factor, and insulin-like growth factor cytokines are essential for tendon repair [23,56,57]. Moreover, tendons are repaired by tenocytes and fibroblasts [58]. Recently, low-temperature He plasma has been reported to induce the proliferation and migration of human fibroblast primary cells through intracellular ROS formation [15]. Therefore, NTAPP irradiation might induce fibroblast proliferation and migration, increase the production of cytokines involved in tendon repair, and improve Achilles tendon healing. However, these hypotheses require further verification.

The limitations of this study include the short observation period of up to six weeks; therefore, a longer observation period should be implemented in future studies. Second, the sample size of this study was small. Third, because rat Achilles tendons were used, their anatomical and regenerative characteristics may differ from those of humans, and future studies on large animals are necessary. However, this *in vivo* study showed that tendon healing was promoted by NTAPP irradiation, which has clinical application potential.

In conclusion, this study shows early tissue repair and the conversion of collagen type III to type I were accelerated by the NTAPP irradiation of the Achilles tendons.

According to the mechanical testing results, the load to failure increased at an early stage. These results suggest that NTAPP may improve Achilles tendon repair. Further research is required to elucidate these mechanisms.

The clinical application of NTAPP's restorative effect on Achilles tendon injuries is expected to shorten the bracing period and enable an earlier return to sports. It may also prove effective for the treatment of other tendon injuries, and is expected to be explored further in this context in the future.

## Supporting information

**S1 Data.**
(XLSX)

## Acknowledgments

We appreciate contributions from all group members in the laboratory of Osaka City University.

## Author Contributions

**Conceptualization:** Katusmasa Nakazawa, Hiromitsu Toyoda, Tomoya Manaka, Tatsuru Shirafuji, Hiroaki Nakamura.

**Data curation:** Katusmasa Nakazawa, Hiromitsu Toyoda, Tomoya Manaka, Kumi Orita, Yoshihiro Hirakawa, Kosuke Saito, Ryosuke Iio, Akiyoshi Shimatani, Yoshitaka Ban, Hana Yao, Ryosuke Otsuki, Yamato Torii.

**Formal analysis:** Katusmasa Nakazawa, Kumi Orita, Yoshihiro Hirakawa, Kosuke Saito, Ryosuke Iio, Akiyoshi Shimatani, Yoshitaka Ban, Hana Yao, Yamato Torii, Jun-Seok Oh.

**Funding acquisition:** Hiromitsu Toyoda, Jun-Seok Oh, Tatsuru Shirafuji, Hiroaki Nakamura.

**Investigation:** Katusmasa Nakazawa, Hiromitsu Toyoda, Tomoya Manaka, Jun-Seok Oh, Tatsuru Shirafuji.

**Methodology:** Katusmasa Nakazawa, Hiromitsu Toyoda, Tomoya Manaka, Hiroaki Nakamura.

**Project administration:** Hiromitsu Toyoda, Tomoya Manaka, Tatsuru Shirafuji, Hiroaki Nakamura.

**Supervision:** Hiromitsu Toyoda, Tomoya Manaka, Jun-Seok Oh, Tatsuru Shirafuji, Hiroaki Nakamura.

**Validation:** Katusmasa Nakazawa, Hiromitsu Toyoda, Tomoya Manaka, Kumi Orita, Yoshihiro Hirakawa, Kosuke Saito, Ryosuke Iio, Akiyoshi Shimatani, Yoshitaka Ban, Hana Yao, Ryosuke Otsuki, Yamato Torii, Jun-Seok Oh.

**Visualization:** Katusmasa Nakazawa, Yoshihiro Hirakawa, Kosuke Saito, Ryosuke Iio, Akiyoshi Shimatani, Yoshitaka Ban, Hana Yao, Ryosuke Otsuki, Yamato Torii, Jun-Seok Oh.

**Writing – original draft:** Katusmasa Nakazawa, Hiromitsu Toyoda, Tomoya Manaka, Kumi Orita, Jun-Seok Oh.

**Writing – review & editing:** Katusmasa Nakazawa, Hiromitsu Toyoda, Tomoya Manaka, Jun-Seok Oh, Tatsuru Shirafuji, Hiroaki Nakamura.

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
