## [Decision Letter · Decision Letter 0]

13 Oct 2023

PONE-D-23-17753In vivo study on the repair of Achilles tendon injury treated with non-thermal atmospheric-pressure helium microplasma jetPLOS ONE

Dear Dr. Toyoda,

Thank you for submitting your manuscript to PLOS ONE. After careful consideration, we feel that it has merit but does not fully meet PLOS ONE’s publication criteria as it currently stands. Therefore, we invite you to submit a revised version of the manuscript that addresses the points raised during the review process.

We look forward to receiving your revised manuscript.

Kind regards,

Carlos Alberto Antunes Viegas, DVM; MSc; PhD

Academic Editor

PLOS ONE

Journal Requirements:

Reviewers' comments:

Reviewer's Responses to Questions

**Comments to the Author**

1. Is the manuscript technically sound, and do the data support the conclusions?

Reviewer #1: Yes

Reviewer #2: Partly

2. Has the statistical analysis been performed appropriately and rigorously? 

Reviewer #1: Yes

Reviewer #2: Yes

3. Have the authors made all data underlying the findings in their manuscript fully available?

Reviewer #1: Yes

Reviewer #2: No

4. Is the manuscript presented in an intelligible fashion and written in standard English?

Reviewer #1: Yes

Reviewer #2: Yes

5. Review Comments to the Author

Reviewer #1: General comments:

This is a very interesting manuscript, very well written and structured. The topic is of scientific infesters and with possible e future application in a clinical point of view.

Specific comments:

- The title should include some reference to this population, thus the rat as a model; Also this should be clearer in the abstract.

- Line 79: After the objectivo, I suggest to add the hypothesis of this study;

- Line 146: When the authors refer to the histological evaluation by three authors. Do you mean blinded evaluation? Please specify this and how was the method of evaluation.

- Results section: In this section there are some sentences that seems to discuss the results and should be on the discussion section and not here. For example: Line 247: "Thus, the appearance of type I and II ..."; Line 252: " This indicates that the collagen networks ..."; Line 263: " This clearly indicated early..."; Line 279: " These result show that NTAPP irradiation ...."Line 289: " thus plasma-treated improved...". All these are discussion or conclusions and should not be here.

- Discussion section: This section is very well written and I would only suggest to add some more bibliographic reference. For example: Line 303 when talking about scar tissue formation; line 321 when talking about achilles tendon maturation; line 352 when talking about the reports that have been published.

- Paragraph 342-351: It is confusing I suggest to explain better.

- Paragraph 360- 365: Once again the hypothesis should be first mentioned after the objective and then I also suggest to re-write this part for better understanding.

- Line 372: Replace "here" for something more specific as " in this study" or " with this population" or " in this case", etc.

- In the last paragraph I suggest to make a point on how this could be important for future application in the clinical setting.

Reviewer #2: In their paper “In vivo study on the repair of Achilles tendon injury treated with non-thermal atmospheric-pressure helium microplasma jet” Nakazawa and co-worker have tenotomized rats. Afterwards they have resuturerd the Achilles’tendon and applied helium plasma to one group. The healing success was investigated using histology, biomechanics and XPS. The set-up appeared to be sound. There appear to be an effect of the treatment, which is significantly measurable. However even I value Gundula Schulze-Tanzil and her group greatly, I do not agree to put all histomorphometric parameters into one score. Especially, when experiencing different issues. On the one hand there is enchondral ossification (due to tendon degeneration) which is detectable by alzian blue staining. However, this staining will disappear over time when the degenerated tendon goes into ossification. On the other hand there is scar formation, which is found by several other parameters (like cellularity). Putting these two things in the same basket is not the same in my point of view. The quantitative analysis of collagen I is unfortunately completely unacceptable in this form. Picrosirius staining with polarization microscopy is not a way to quantify the collagen I content in tendons. The color shift is an indicator for fibril thickness but not for a change in protein. Collagen type I can be of every color in picro-sirus staining depending on the collagen maturation stage and collagen type V content. Speaking of collagen type V- please reinvestigate line 242-248 because there are many things which are not right like the role of collagen V. Also there is a gradient of collagen I to collagen III dependent on the tissue and stage. Skin has quite a big portion of collagen type I besides collagen type III. In tendon it is inversed. The minor fibrillar collagens are not correctly mentioned too.

Please give us a graphical representation of the results of the biomechanics and picro-sirius quantification, there are better to internalize than the numbers inside the text.

The discussion is not sound since you have overinterpreted the picro-sirius data. To draw these conclusions you need other methods, like mass spectrometry or antibody driven detection methods (immune fluorescence or western blotting).

Please address these issues prior to acceptance of the paper, since the concept and the outcome of the treatment is quite convincing.

6. PLOS authors have the option to publish the peer review history of their article (what does this mean?). If published, this will include your full peer review and any attached files.

Reviewer #1: No

Reviewer #2: **Yes: **Daniel Kronenberg

---

## [Author Response · Author response to Decision Letter 0]

24 Nov 2023

Reviewer #1: General comments:

This is a very interesting manuscript, very well written and structured. The topic is of scientific infesters and with possible e future application in a clinical point of view.

Specific comments:

- The title should include some reference to this population, thus the rat as a model; Also this should be clearer in the abstract.

RESPONSE: Thank you for pointing this out. We have revised the title to “In vivo study on the repair of Rat Achilles tendon injury treated with non-thermal atmospheric-pressure helium microplasma jet ”.

- Line 79: After the objective, I suggest to add the hypothesis of this study;

RESPONSE: Thank you for this suggestion. We have revised the final paragraph as follows:

"We hypothesize that NTAPP irradiation of the Achilles tendon enhances the cell proliferative capacity and promote early Achilles tendon repair. ” in Lines 80–82.

- Line 146: When the authors refer to the histological evaluation by three authors. Do you mean blinded evaluation? Please specify this and how was the method of evaluation.

RESPONSE: Thank you for raising this point. The following text has been added to clarify this aspect: “Histological evaluation was performed by three investigators (n = 6) blinded to the treatment and number of weeks.” in Lines 148-149

- Results section: In this section there are some sentences that seems to discuss the results and should be on the discussion section and not here. For example: Line 247: "Thus, the appearance of type I and II ..."; Line 252: " This indicates that the collagen networks ..."; Line 263: " This clearly indicated early..."; Line 279: " These result show that NTAPP irradiation ...."Line 289: " thus plasma-treated improved...". All these are discussion or conclusions and should not be here.

RESPONSE: Thank you for pointing this out, we have moved this content from the Results to the Discussion.

- Discussion section: This section is very well written and I would only suggest to add some more bibliographic reference. For example: Line 303 when talking about scar tissue formation; line 321 when talking about achilles tendon maturation; line 352 when talking about the reports that have been published.

RESPONSE: Thank you for this suggestion. We have added the following references.

37) Schulze-Tanzil GG, Cáceres MD, Stange R, Wildemann B, Docheva D. Tendon healing: a concise review on cellular and molecular mechanisms with a particular focus on the Achilles tendon. Bone Jt Res. 2022;11: 561–574. doi:10.1302/2046-3758.118.BJR-2021-0576.R1

39) Thomopoulos S, Soslowsky LJ, Flanagan CL, Tun S, Keefer CC, Mastaw J, et al. The effect of fibrin clot on healing rat supraspinatus tendon defects. J Shoulder Elb Surg. 2002;11: 239–247. doi:10.1067/mse.2002.122228

40) Aro AA, Simões GF, Esquisatto MAM, Foglio MA, Carvalho JE, Oliveira ALR, et al. Arrabidaea chica extract improves gait recovery and changes collagen content during healing of the Achilles tendon. Injury. 2013;44: 884–892. doi:10.1016/j.injury.2012.08.055

41) Sharma P, Maffulli N. Tendon injury and tendinopathy: Healing and repair. J Bone Jt Surg. 2005;87: 187–202. doi:10.2106/JBJS.D.01850

42)Hou Y, Mao ZB, Wei XL, Lin L, Chen LX, Wang HJ, et al. The roles of TGF-β1 gene transfer on collagen formation during Achilles tendon healing. Biochem Biophys Res Commun. 2009;383: 235–239. doi:10.1016/j.bbrc.2009.03.159

- Paragraph 342-351: It is confusing I suggest to explain better.

RESPONSE: Thank you for this suggestion. The following text has been revised on Lines 354–361:

“NTAPP produces ROS and RNS, including nitrogen monoxide (NO) and nitrogen dioxide (NO2), by interacting with the surrounding atmosphere when gases are irradiated from the plasma jet [51]. Only recently has the role of ROS in the maintenance of normal biological functions become clear [52]. ROS activate intracellular signaling pathways, resulting in the production of soluble factors involved in cell growth and proliferation [53,54]. According to Brun et al., He-generated plasma treatment induces the proliferation and migration of primary human fibroblast-like cells, primarily through intracellular ROS formation [15]. Accordingly, we speculate that the RONS generated through the plasma jet may promote tendon regeneration.”

- Paragraph 360- 365: Once again the hypothesis should be first mentioned after the objective and then I also suggest to re-write this part for better understanding.

RESPONSE: Thank you for this comment. The following text has been revsied on Lines 370–372:

"Therefore, NTAPP irradiation might induce fibroblast proliferation and migration, increase the production of cytokines involved in tendon repair, and improve Achilles tendon healing. However, these hypotheses require further verification.”

- Line 372: Replace "here" for something more specific as " in this study" or " with this population" or " in this case", etc.

RESPONSE: Thank you for these suggestions. The following text has been revised on Lines 383–384:

“In conclusion, this study shows early tissue repair and the conversion of collagen type III to type I were accelerated by the NTAPP irradiation of the Achilles tendons.”

- In the last paragraph I suggest to make a point on how this could be important for future application in the clinical setting.

RESPONSE: Thank you for this suggestion. We have added a brief summary of potential future clinical applications in the final paragraph as follows:

“The clinical application of NTAPP's restorative effect on Achilles tendon injuries is expected to shorten the bracing period and enable an earlier return to sports. It may also prove effective for the treatment of other tendon injuries, and is expected to be explored further in this context in the future.” (Lines 388–390)

 

Reviewer #2: In their paper “In vivo study on the repair of Achilles tendon injury treated with non-thermal atmospheric-pressure helium microplasma jet” Nakazawa and co-worker have tenotomized rats. Afterwards they have resuturerd the Achilles’tendon and applied helium plasma to one group. The healing success was investigated using histology, biomechanics and XPS. The set-up appeared to be sound. There appear to be an effect of the treatment, which is significantly measurable. However even I value Gundula Schulze-Tanzil and her group greatly, I do not agree to put all histomorphometric parameters into one score. Especially, when experiencing different issues. On the one hand there is enchondral ossification (due to tendon degeneration) which is detectable by alzian blue staining. However, this staining will disappear over time when the degenerated tendon goes into ossification. On the other hand there is scar formation, which is found by several other parameters (like cellularity). Putting these two things in the same basket is not the same in my point of view. 

RESPONSE: Thank you very much for your suggestions. We have conducted a statistical review again for each item. Regarding the text, we have added the following on Lines 239–242 and added Table 2:

“The item-by-item comparison revealed that there was no significant difference between items at 2 weeks. However, at 4 weeks, significant differences were observed in cell alignment, cell nucleus morphology, and vascularization in the defect area. At 6 weeks, a significant difference occurrent in the transition from defect to normal tissue." 

We have also included added a description of endochondral ossification on Lines 320–322 as follows:

“In addition, Alcian blue staining revealed cartilage formation starting 2 weeks post-operatively. These cartilage masses reportedly calcify after 4 to 6 weeks and are thought to be caused by endochondral ossification [44].”

The quantitative analysis of collagen I is unfortunately completely unacceptable in this form. Picrosirius staining with polarization microscopy is not a way to quantify the collagen I content in tendons. The color shift is an indicator for fibril thickness but not for a change in protein. Collagen type I can be of every color in picro-sirus staining depending on the collagen maturation stage and collagen type V content. Speaking of collagen type V- please reinvestigate line 242-248 because there are many things which are not right like the role of collagen V. Also there is a gradient of collagen I to collagen III dependent on the tissue and stage. Skin has quite a big portion of collagen type I besides collagen type III. In tendon it is inversed. The minor fibrillar collagens are not correctly mentioned too.

RESPONSE: Thank you very much for your suggestion. The role of type V collagen and the contents regarding Picrosirius red staining were added to Lines 327–332 as follows: 

“Normal Achilles tendon dry weight is composed primarily of collagen type I (90%) [45,46]. However, it also contains type III collagen, present in the fibrocartilage of tendons and tendons, as well as type V collagen, which forms the nuclei type I collagen protofibrils and participates in fiber diameter regulation [47]. Picrosirius red staining detects and quantitatively evaluates collagen in normal and overlying tissue sections and can effectively identify collagen type I and collagen type III in the musculoskeletal system, liver, gastrointestinal tract, skin, and myocardium [48].”

As you noted, there was a possibility of type V collagen involvement. Accordingly we have included the following as a study limitation: 

“Fourth, the quantitative evaluation of type I collagen content via picrosirius red staining may be affected by the maturation stage of collagen and the type V collagen content, which must be considered when interpreting the results." (Lines 378–380)

Please give us a graphical representation of the results of the biomechanics and picro-sirius quantification, there are better to internalize than the numbers inside the text.

The discussion is not sound since you have overinterpreted the picro-sirius data. To draw these conclusions you need other methods, like mass spectrometry or antibody driven detection methods (immune fluorescence or western blotting).

RESPONSE: We have addressed this comment as a study limitation on Lines 380–382 as follows: 

“Hence, mass spectrometry analyses or antibody detection methods (immunofluorescence or western blotting) are warranted to address these possibilities.”

We have also added the biomechanics and picro-sirius quantification results to Figures 10 and 11.

---

## [Decision Letter · Decision Letter 1]

14 Feb 2024

PONE-D-23-17753R1In vivo study on the repair of Rat Achilles tendon injury treated with non-thermal atmospheric-pressure helium microplasma jetPLOS ONE

Dear Dr. Toyoda,

Thank you for submitting your manuscript to PLOS ONE. After careful consideration, we feel that it has merit but does not fully meet PLOS ONE’s publication criteria as it currently stands. Therefore, we invite you to submit a revised version of the manuscript that addresses the points raised during the review process. Please take into account the response to the items requested by reviewer 2 and submit your revised manuscript by Mar 30 2024 11:59PM. If you will need more time than this to complete your revisions, please reply to this message or contact the journal office at plosone@plos.org. Please include the following items when submitting your revised manuscript:A rebuttal letter that responds to each point raised by the academic editor and reviewer(s). You should upload this letter as a separate file labeled 'Response to Reviewers'.A marked-up copy of your manuscript that highlights changes made to the original version. You should upload this as a separate file labeled 'Revised Manuscript with Track Changes'.An unmarked version of your revised paper without tracked changes. You should upload this as a separate file labeled 'Manuscript'.

We look forward to receiving your revised manuscript.

Kind regards,

Carlos Alberto Antunes Viegas, DVM; MSc; PhD

Academic Editor

PLOS ONE

Reviewers' comments:

Reviewer's Responses to Questions

**Comments to the Author**

1. If the authors have adequately addressed your comments raised in a previous round of review and you feel that this manuscript is now acceptable for publication, you may indicate that here to bypass the “Comments to the Author” section, enter your conflict of interest statement in the “Confidential to Editor” section, and submit your "Accept" recommendation.

Reviewer #1: All comments have been addressed

Reviewer #2: (No Response)

2. Is the manuscript technically sound, and do the data support the conclusions?

Reviewer #1: Yes

Reviewer #2: Yes

3. Has the statistical analysis been performed appropriately and rigorously? 

Reviewer #1: Yes

Reviewer #2: Yes

4. Have the authors made all data underlying the findings in their manuscript fully available?

Reviewer #1: Yes

Reviewer #2: Yes

5. Is the manuscript presented in an intelligible fashion and written in standard English?

Reviewer #1: Yes

Reviewer #2: Yes

6. Review Comments to the Author

Reviewer #1: This manuscript is well structured and debates an important theme for the clinical practice. Although it is performed in rats, it has the potential to be translated for dogs and cats. I think this study should be continued and try to apply in dogs and cats.

The authors coped with every proposed alteration, and thus it is acceptable for publication.

Reviewer #2: I still think that the authors should add an additional way to quantify the collagen content. The authors have prepared the histological slides and can quantify collagen IHC.

7. PLOS authors have the option to publish the peer review history of their article (what does this mean?). If published, this will include your full peer review and any attached files.

Reviewer #1: **Yes: **Angela Martins

Reviewer #2: **Yes: **Daniel Kronenberg

---

## [Author Response · Author response to Decision Letter 1]

9 Mar 2024

Reviewer #1: General comments:

This is a very interesting manuscript, very well written and structured. The topic is of scientific infesters and with possible e future application in a clinical point of view.

Specific comments:

- The title should include some reference to this population, thus the rat as a model; Also this should be clearer in the abstract.

RESPONSE: Thank you for pointing this out. We have revised the title to “In vivo study on the repair of Rat Achilles tendon injury treated with non-thermal atmospheric-pressure helium microplasma jet ”.

Reviewer #1: This manuscript is well structured and debates an important theme for the clinical practice. Although it is performed in rats, it has the potential to be translated for dogs and cats. I think this study should be continued and try to apply in dogs and cats.

The authors coped with every proposed alteration, and thus it is acceptable for publication.

RESPONSE: Thank you very much for your evaluation. We would like to study the effects of low-temperature atmospheric pressure plasma using large animals in the future.

Reviewer #2: I still think that the authors should add an additional way to quantify the collagen content. The authors have prepared the histological slides and can quantify collagen IHC.

RESPONSE: Thank you for pointing this out. We have now added Immunohistochemical testing for collagen type Ⅰ and Ⅲ. Collagen type Ⅰ was increased in the plasma treatment group during the entire period, and collagen Ⅲ was decreased at 2 and early 4 weeks.

The above results are noted in the text as follows;

Line 167-179 

“Immunohistochemical testing

Paraffin-embedded sections were deparaffinized using xylene and dehydrated through graded alcohols. Slides were pretreated with citrate buffer (Target Retrieval Solution [S1699], 10×; DAKO Japan, Tokyo, Japan) in phosphate-buffered saline solution (PBS) for 20 minutes at 90°C for optimal antigen retrieval. Endogenous peroxidases were quenched using 1.0% hydrogen peroxidase in methanol for 30 minutes at room temperature. Slides were then rinsed with PBS and incubated with 10% goat serum for 30 minutes at room temperature. Subsequently, specimens were incubated with rabbit primary antibodies (type I collagen, 1:100 dilution; Abcam, ab34710 and type III collagen, 1:100 dilution; Abcam, ab6310) at 4°C overnight. After extensive washing with PBS, slides were incubated with a peroxidase-labeled antibody (Histofine Simple Stain; Nichirei Biosciences, Tokyo, Japan) for 30 minutes at room temperature. After extensive washing with PBS, the immunoreaction was visualized by incubating the sections for 3 minutes in 3,3′-diaminobenzidine (Histofine Simple DAB solution; Nichirei Biosciences).”

Line 283-306 

“Quantitative analysis of the type I and III collagen content via immunohistochemistry testing

Type I collagen expression was observed in the plasma-treated group from 2 weeks and became stronger with time, showing stronger staining compared to the untreated control group during the entire period (Fig 11 A-F). Type III collagen expression was stronger in the untreated control group than in the plasma-treated group at 2 and 4 weeks (Fig 12 A, B, D, and E), but the staining was similar at 6 weeks (Fig 12 C and F). Expression decreased over time in both groups. In quantitative analysis, type I collagen expression was significantly higher in the plasma-treated group at 2, 4, and 6 weeks. (p = 0.010, p = 0.025, and p = 0.025, respectively). Type III collagen expression was significantly lower in the plasma-treated group at 2 and 4 weeks (p = 0.010 and p = 0.037), but there was no significant difference between the two groups at 6 weeks. (p = 0.262) (Fig 13)

Fig 11. Immunohistochemical testing against type I collagen results. (A–C) Tissue after He microplasma jet treatment at 2, 4, and 6 weeks post-surgery, respectively. (D–F) Untreated control tissues at 2, 4, and 6 weeks post-surgery, respectively. Scale bar, 200 µm (originally 100 µm).

Fig 12. Immunohistochemical testing against type　Ⅲ collagen results. (A–C) Tissue after He microplasma jet treatment at 2, 4, and 6 weeks post-surgery, respectively. (D–F) Untreated control tissues at 2, 4, and 6 weeks post-surgery, respectively. Scale bar, 200 µm (originally 100 µm).

Fig 13. Quantitative analysis of the type I and Ⅲ collagen content via Immunohistochemistry testing 

Type I collagen expression in He microplasma jet-treated group was higher than that in untreated control group in all time point. (a: p = 0.010, b: p = 0.025, c: p = 0.025) Type III collagen expression was significantly lower in the He microplasma jet-treated group at 2 and 4 weeks. (a: p = 0.010, b: p = 0.037)”

We have also added the immunohistochemical testing results to Figures 11, 12 and 13.

Fig. 13 is changed to 14 on Line 294-306.

Discussion also edits as follows.

Line 338-339 

“Immunochemical staining results also showed more type 1 collagen in the plasma-treated group and less type 3 collagen in the plasma-treated group at 2 and 4 weeks.”

Line 375-378 

“Here, the type I collagen content also resulted in higher brightness in Picrosirius red staining and the expression of type I collagen in immunochemical stainning at 2, 4, and 6 weeks post-surgery. The expression of type III collagen was also decreased in the plasma treatment group at 2 and 4 weeks.”

---

## [Decision Letter · Decision Letter 2]

12 Mar 2024

In vivo study on the repair of Rat Achilles tendon injury treated with non-thermal atmospheric-pressure helium microplasma jet

PONE-D-23-17753R2

Dear Dr. Hiromitsu Toyoda,

We’re pleased to inform you that your manuscript has been judged scientifically suitable for publication and will be formally accepted for publication once it meets all outstanding technical requirements.

Kind regards,

Carlos Alberto Antunes Viegas, DVM; MSc; PhD

Academic Editor

PLOS ONE

Additional Editor Comments (optional):

Reviewers' comments:

Reviewer's Responses to Questions

**Comments to the Author**

1. If the authors have adequately addressed your comments raised in a previous round of review and you feel that this manuscript is now acceptable for publication, you may indicate that here to bypass the “Comments to the Author” section, enter your conflict of interest statement in the “Confidential to Editor” section, and submit your "Accept" recommendation.

Reviewer #2: All comments have been addressed

2. Is the manuscript technically sound, and do the data support the conclusions?

Reviewer #2: Yes

3. Has the statistical analysis been performed appropriately and rigorously? 

Reviewer #2: Yes

4. Have the authors made all data underlying the findings in their manuscript fully available?

Reviewer #2: (No Response)

5. Is the manuscript presented in an intelligible fashion and written in standard English?

Reviewer #2: Yes

6. Review Comments to the Author

Reviewer #2: With adding additional histological experiments i.e. staining for collagen type I and III and the corresponding quantification, the authors met all my remaining comments

Thank you

7. PLOS authors have the option to publish the peer review history of their article (what does this mean?). If published, this will include your full peer review and any attached files.

Reviewer #2: **Yes: **Daniel Kronenberg

---

## [Editor Report · Acceptance letter]

14 Mar 2024

PONE-D-23-17753R2 

PLOS ONE

Dear Dr. Toyoda, 

I'm pleased to inform you that your manuscript has been deemed suitable for publication in PLOS ONE. Congratulations! Your manuscript is now being handed over to our production team.

Kind regards, 

on behalf of

Dr. Carlos Alberto Antunes Viegas 

Academic Editor

PLOS ONE